# DDX5 inhibits inflammation by modulating m6A levels of TLR2/4 transcripts during bacterial infection

Jian Xu [1,6], Li-Yuan Liu[1,6], Fei-Jie Zhi[1], Yin-Juan Song[1], Zi-Hui Zhang[2], Bin Li[3], Fu-Ying Zheng[1], Peng-Cheng Gao[1], Su-Zi Zhang[1], Yu-Yu Zhang[1,3], Ying Zhang[1], Ying Qiu[4], Bo Jiang[5], Yong-Qing Li[5], Chen Peng [2✉] & Yue-Feng Chu [1✉]

## Abstract

**DExD/H-box helicases are crucial regulators of RNA metabolism and antiviral innate immune responses; however, their role in bacteria-induced inflammation remains unclear. Here, we report that DDX5 interacts with METTL3 and METTL14 to form an m6A writing complex, which adds N6-methyladenosine to transcripts of toll-like receptor (TLR) 2 and TLR4, promoting their decay via YTHDF2-mediated RNA degradation, resulting in reduced expression of TLR2/4. Upon bacterial infection, DDX5 is recruited to Hrd1 at the endoplasmic reticulum in an MyD88-dependent manner and is degraded by the ubiquitin-proteasome pathway. This process disrupts the DDX5 m6A writing complex and halts m6A modification as well as degradation of TLR2/4 mRNAs, thereby promoting the expression of TLR2 and TLR4 and downstream NF-κB activation. The role of DDX5 in regulating inflammation is also validated in vivo, as DDX5- and METTL3-KO mice exhibit enhanced expression of inflammatory cytokines. Our findings show that DDX5 acts as a molecular switch to regulate inflammation during bacterial infection and shed light on mechanisms of quiescent inflammation during homeostasis.**

**Keywords** DDX5; N6-Methyladenosine; TLR2/4 Transcripts; Inflammation; Bacterial Infection
**Subject Categories** Microbiology, Virology & Host Pathogen Interaction; RNA Biology; Signal Transduction

## Introduction

DExD/H-box (DDX or DHX) helicases are common RNA-binding proteins that participate in RNA metabolism, transcription, alternative pre-mRNA splicing, mRNA nuclear export, and mRNA decay (Linder and Jankowsky, 2011; De Bortoli et al, 2021; Sloan and Bohnsack, 2018). They control gene transcription and translation of mRNA transcripts by affecting mRNA stability, splicing, and the initiation of translation. Unsurprisingly, they have emerged as essential regulators of various biological processes (Fuller-Pace, 2006; Shen and Pelletier, 2020), including antiviral innate immunity (Tang and Zheng, 2021; Baldaccini and Pfeffer, 2021; Ullah et al, 2022). Specifically, DDX1, DDX21, and DHX36 form a viral sensor complex that regulates the Toll-like receptor (TLR) 3/7-mediated TRIF pathway and triggers the innate immune response (Zhang et al, 2011). DHX15, DDX60, and DHX29 regulate the RIG-I/MDA5-mediated type I interferon signaling pathway (Mosallanejad et al, 2014; Oshiumi et al, 2015; Zhu et al, 2018). Furthermore, DDX41, DDX3, and DDX19 interact with DNA and regulate the Tank-binding kinase 1-interferon regulatory transcription factor (TBK1-IRF) signaling pathway (Soulat et al, 2008; Zhang et al, 2019; Singh et al, 2022). Moreover, DHX15 utilizes NOD-like receptor family pyrin domain containing 6 (NLRP6) to regulate the mitochondrial antiviral signaling protein (MAVS)/IRF3 signaling pathway (Xing et al, 2021), and DDX17 has been identified as an inflammasome sensor for retrotransposon RNAs (Minton, 2022). Certain DExD/H-box helicases, such as DHX9, DDX17, and DDX19, have been associated with the production of inflammatory cytokines during viral infection (Zhang et al, 2019; Dempsey et al, 2018; Minton, 2022); however, none of them has been found to control the inflammatory response during bacterial infection.

DDX5 is the most abundant RNA helicase and is aberrantly expressed in most malignant tumors, where it participates in transcriptional regulation, differentiation, metastasis, and

[1]State Key Laboratory for Animal Disease Control and Prevention, College of Veterinary Medicine, Lanzhou University, Lanzhou Veterinary Research Institute, Chinese Academy of Agricultural Sciences, Lanzhou, China. [2]National Key Laboratory of Veterinary Public Health, College of Veterinary Medicine, China Agricultural University, Beijing, China. [3]College of Veterinary Medicine, Xinjiang Agricultural University, Urumqi, China. [4]State Key Laboratory of Cell Biology, Shanghai Institute of Biochemistry and Cell Biology, Center for Excellence in Molecular Cell Science, Chinese Academy of Sciences, University of Chinese Academy of Sciences, Shanghai, China. [5]Institue of Animal Husbandry and Veterinary Medicine, Beijing Academy of Agriculture and Forestry Sciences, Beijing, China. [6]These authors contributed equally: Jian Xu, Li-Yuan Liu.
✉E-mail: pengchenea@cau.edu.cn; chuyuefeng@caas.cn

regulation of various signaling pathways (Nyamao et al, 2019). DDX5 is also involved in cellular differentiation, embryonic development, metabolism, as well as viral infection (Cheng et al, 2018). Interestingly, DDX5 could act as a positive regulator of replication for the *Japanese encephalitis virus*, *influenza virus*, and severe acute respiratory syndrome coronavirus-2 (*SARS-CoV-2*) but a negative regulator for *Myxoma virus* and *hepatitis B virus* (Cheng et al, 2018). Our previous study demonstrated that DDX5 was hijacked by an avian oncogenic herpesvirus to inhibit interferon-beta production, thereby promoting viral replication (Xu et al, 2021a). Moreover, DDX5 promotes infection of RNA viruses by regulating N6-methyladenosine (m6A) levels on DHX58 and nuclear factor kappa beta (NF-κB) transcripts to dampen antiviral innate immunity (Xu et al, 2021b; Zhao et al, 2022). Although the latest studies showed that IL-17D modulation of DDX5 expression controls inflammation in keratinocytes during inflammatory skin diseases (Ni et al, 2022) and that DDX5 regulates intestinal CD4+ T cell population heterogeneity, as well as cytokine production capacities in a context-dependent manner (Ma et al, 2023). However, no studies so far have examined whether RNA helicases contribute to bacterial inflammation or the molecular basis of inflammation quiescence during homeostasis. The m6A modification is a crucial step for the modulation of inflammation and maintaining homeostasis (Luo et al, 2021), and the m6A demethylase ALKBH5 is required for antibacterial innate defense via intrinsic activation of neutrophil migration (Liu et al, 2022). These findings suggested the possible role of mRNA m6A modification in regulating bacteria-induced inflammation. As viruses and bacteria induce inflammatory responses through activation of distinct pattern recognition receptors (PRRs), it is worth asking if bacteria-induced inflammation was also impacted by DDX5-mediated m6A modification.

TLR2 and TLR4 are the most critical PRRs during bacterial infection and function to promote inflammatory responses upon invasion of bacteria. Activation of TLR2/4 can induce NF-κB-regulated inflammatory cytokines, such as interleukin 6 (IL-6) and tumor necrosis factor-alpha (TNF-α) during bacterial infection (Meizlish et al, 2021). Unfortunately, transcriptional regulation of TLR2/4 has remained unclear. In the present study, we investigated the biological functions of DDX5 during bacterial infection with a focus on its role in bacteria-promoted inflammation. We discovered that under normal conditions (i.e., without bacterial infection), DDX5 interacts with two known m6A writer proteins, METTL3 and METTL14, to form a ternary m6A writer complex, which recognizes and binds to TLR2/4 mRNAs to introduce m6A modifications. The resulting m6A-modified mRNAs are susceptible to Y521-B homology domain family 2 (YTHDF2)-mediated mRNA degradation, leading to reduced TLR2/4 expression on the cell membrane and inactivation of NF-κB-mediated synthesis of inflammatory cytokines. Conversely, following bacterial infection, DDX5 was targeted by Hrd1, an ER-localized E3 ligase (Thepsuwan et al, 2023), and subsequently degraded. Degradation of DDX5 prohibits the formation of the DDX5-METTL3-METTL14 complex, thereby reducing m6A modification levels of TLR2/4 mRNAs, resulting in the restoration of membrane expression of TLR2/4 and the induction of NF-κB-mediated inflammation pathways. Thus, our study not only sheds light on the regulatory function of DDX5 in inflammation following bacterial infection but also reveals a novel mechanism by which inflammation homeostasis is maintained.

# Results

## DDX5 blocks the synthesis of IL-6 and TNF-α during bacterial infection

To evaluate the role of DDX5 during bacterial infection, we first examined the effect of DDX5 on inflammatory responses in vivo. Homozygous DDX5$^{-/-}$ mice suffered from embryonic lethality or infertility (Fan et al, 2017). However, as shown in our previous study, there was no difference in terms of physiological functions between heterozygous DDX5$^{+/-}$ mice and wild-type DDX5$^{+/+}$ mice, except for the decreased DDX5 synthesis in the lungs of DDX5$^{+/-}$ mice (Xu et al, 2021b). We investigated the levels of inflammatory cytokines in serum and tissue of DDX5$^{+/+}$ and DDX5$^{+/-}$ mice in vivo upon infection with *Mycoplasma pneumonia* (*M. pneumonia*), *Staphylococcus aureus* (*S. aureus*), and *Pasteurella multocida* (*P. multocida*). We observed significantly higher levels of IL-6 and TNF-α in both serum (Fig. 1A,B) and lungs of DDX5$^{+/-}$ mice than that in DDX5$^{+/+}$ mice following infection with *P. multocida* and *S. aureus* for 0–12 h, and *M. pneumoniae* for 0–24 h (Fig. 1C,D). In addition, the enhanced damage caused by excessive inflammatory response in the lungs of bacterial-infected DDX5$^{+/-}$ mice was confirmed by hematoxylin and eosin tissue staining (Fig. 1E). Moreover, we determined the production of IL-6 by ELISA in unmodified mouse embryonic fibroblasts (MEFs), or cells over-expressed DDX5 as well as in DDX5-deficient cells after infected with *M. pneumoniae*, *S. aureus*, and *P. multocida*. Indeed, the level of IL-6 was significantly lowered in DDX5-overexpressing MEFs (Appendix Fig. S1A) and significantly increased in DDX5-deficient MEFs when compared to the controls upon infection (Appendix Fig. S1B). These results showed that DDX5 blocked IL-6 or TNF-α release in cells infected with gram-negative, gram-positive bacteria and mycoplasma. As TLR2 and TLR4 are the major PAMPs for pathogenic bacteria (Meizlish et al, 2021), we decided to stimulate mouse macrophages (Mø) or MEFs with lipopolysaccharides (LPS), a TLR4 agonist, or FSL-1 and Pam3CSK4, two TLR2 agonists and investigated if DDX5 could alter PAMP-stimulated IL-6 or TNF-α synthesis (Hally et al, 2018). Similar to the results of bacterial infection, IL-6 or TNF-α production in response to PAMP ligands was significantly increased in DDX5-deficient Mø (Appendix Fig. S1C,D). As expected, IL-6 was significantly increased in DDX5-deficient MEFs (Appendix Fig. S1E) while decreased in DDX5-overexpressed MEFs (Appendix Fig. S1F) regardless of the inflammatory agonist used. To explore if DDX5 affects IL-6 synthesis by modulating its transcription, we investigated the promoter activity of IL-6 in DDX5-overexpressing MEFs using a luciferase reporter assay, in which the synthesis of luciferase was regulated by a known IL-6 promoter motif. The results showed ectopic expression of DDX5 inhibited the synthesis of the reporter gene, suggesting the inhibitory effect of DDX5 on IL-6 synthesis was on the transcriptional level (Fig. EV1A). Taken together, these data demonstrate that DDX5 blocks IL-6 or TNF-α synthesis during bacterial infection in vitro and in vivo.

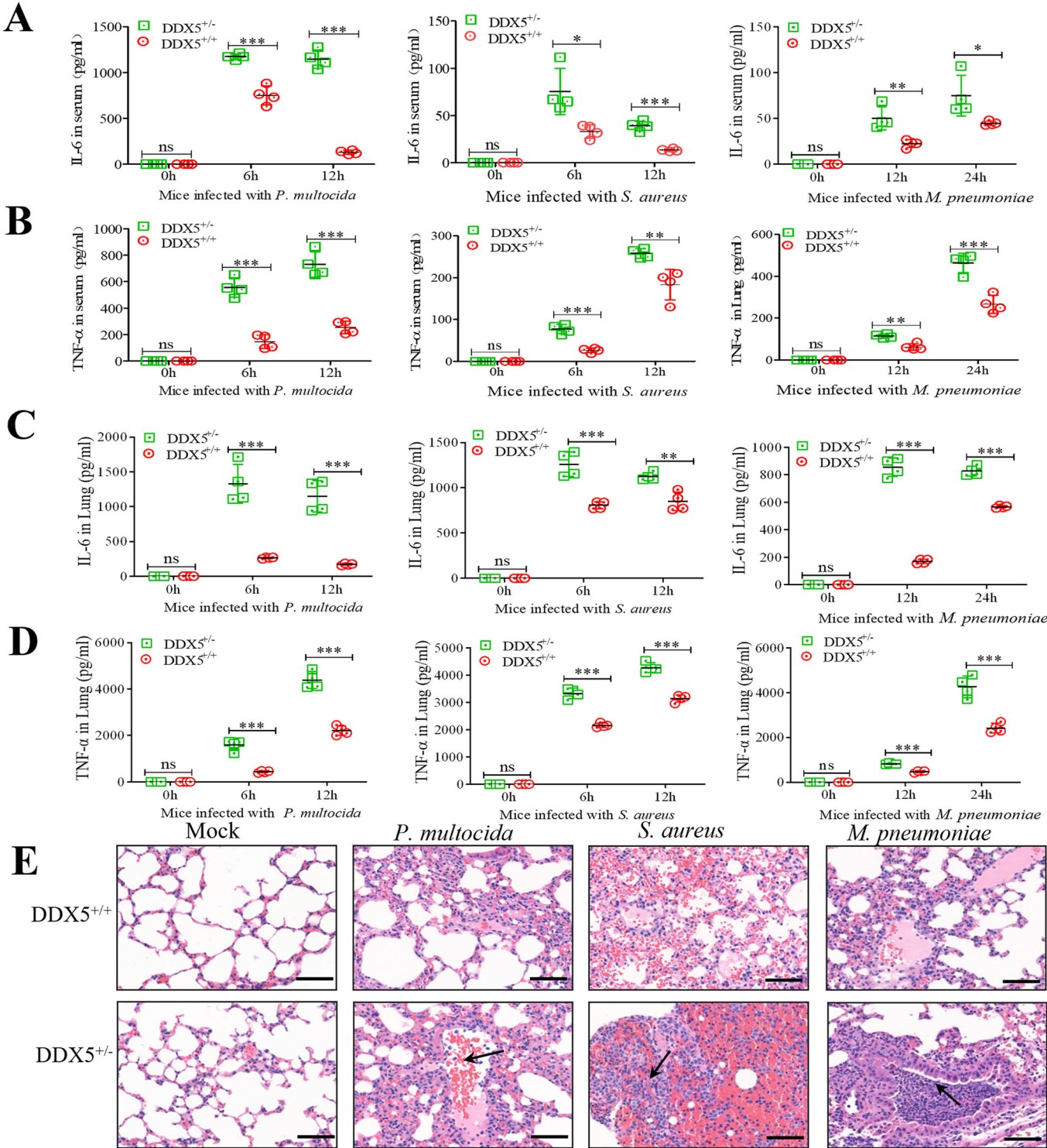

### Bacterial infection induces the degradation of DDX5 by ER-localized Hrd1 via the ubiquitin-proteasome pathway

As our data demonstrated the crucial role of DDX5 in modulating inflammatory responses upon bacterial infection, we next explored how DDX5 was affected by bacterial infection or TLR agonists. We first examined DDX5 levels in Mø and MEFs by Western blotting analysis. DDX5 synthesis exhibited an obvious decrease in MEFs during 0 ~ 6 h post-infection with *S. aureus* and *P. multocida*, as well as during 0-16 h post-infection with *M. pneumoniae* (Fig. 2A). A similar trend was also detected in mouse macrophages (Fig. EV1B). Furthermore, DDX5 protein synthesis was reduced both in MEFs (Fig. 2B) and mouse macrophages (Fig. EV1C) when they were treated with the TLR2/4 agonists LPS, FSL-1 and Pam3CSK4.

◄  **Figure 1.  DDX5 suppresses the production of IL-6 and TNF-α triggered by pathogenic bacteria or TLR2/4 agonists.**

(A, B) IL-6 (A) or TNF-α (B) production in the serum of DDX5$^{+/+}$ or DDX5$^{+/-}$ mice infected with *P. multocida* or *S. aureus* for 6 and 12 h, or those infected with *M. pneumoniae* for 12 and 24 h ($n = 4$). (C, D) IL-6 (C) or TNF-α (D) production in the lungs of DDX5$^{+/+}$ or DDX5$^{+/-}$ mice infected with *P. multocida* or *S. aureus* for 6 and 12 h, or those infected with *M. pneumoniae* for 12 and 24 h ($n = 4$). (E) Hematoxylin and eosin staining of lung tissue from DDX5$^{+/+}$ or DDX5$^{+/-}$ mice infected with *P. multocida* for 12 h, *S. aureus* for 12 h, and *M. pneumoniae* for 24 h. Scale bars, 50 μm. Black arrows indicate severe inflammatory pathology. Data information: In (A–D), all data are represented as the mean ± SEM of three biologically independent samples. ND not detected. "ns" indicates no significant difference, *$p < 0.05$, **$p < 0.01$, and ***$p < 0.001$ (Student's *t* test). Source data are available online for this figure.

These data demonstrate that the levels of DDX5 decrease in response to bacterial infection and upon stimulation with TLR2 and TLR4 agonists.

The decrease of DDX5 observed in MEFs could be the result of protein degradation or blockade of protein synthesis. To determine the nature of DDX5 reduction, MG132, a widely used proteasomal inhibitor was added in cells treated with TLR2/4 agonists. As shown in Figs. 2C and EV1D, addition of MG132 completely rescued the reduction of DDX5 induced by TLR ligands in MEFs and mouse macrophages, indicating the reduction of DDX5 upon TLR activation was due to protein degradation by proteasome. To corroborate this result, we also performed co-immunoprecipitation analysis by pulling down DDX5 in both mouse macrophages and MEFs treated with TLR agonists with an anti-DDX5 antibody and was able to observe phenomenal ubiquitination of DDX5, further confirming the degradation of DDX5 was dependent on K48 linked ubiquitin-proteasome pathway but not the K63 linked ubiquitination (Fig. 2D, Appendix Fig. S2A,B). Previous studies showed Hrd1, an ER-localized E3 ligase was responsible for the degradation of nucleoproteins, such as USP15 in response to bacterial infection and METTL14 to suppress ER proteotoxic liver disease, these nucleoproteins would be exported from the nucleus and degraded by the ER-localized E3 ligase Hrd1 (Lu et al, 2019; Wei et al, 2021). We asked if Hrd1 was the E3 ligase to mediate DDX5 degradation. We collected total proteins in MEFs to pull down Hrd1 with an anti-Hrd1 antibody, followed by mass-spectrometry analysis. In this assay, we found that DDX5 was the primary binding target of Hrd1 (Appendix Fig. S3). Next, we observed that DDX5 was exported from the nucleus after treatment of TLR2/TLR4 agonists in both mouse macrophages (Mø) and MEFs (Fig. 3A and Appendix Fig. S4A). This association was further verified by a co-IP analysis in MEFs co-transfected with plasmids encoding Flag-Hrd1 and Myc-DDX5 (Fig. 3B). Remarkably, depression of Hrd1 in MEFs or mouse macrophages by RNA interference significantly reduced DDX5 degradation triggered by TLR2/4 activation (Appendix Fig. S4B and Fig. 3C). The enzymatic activity of Hrd1 was essential for DDX5 degradation as an Hrd1 mutant (C329S) with inactivated E3 ligase activity failed to promote DDX5 degradation (Fig. 3D). The interaction of DDX5 and Hrd1 was also significantly enhanced in MEFs (Fig. 3E) or macrophages (Appendix Fig. S4C) after stimulation with inflammatory agonists (LPS, FSL-1, and Pam3CSK4), and CLSM exhibited increased co-localization with DDX5 and Hrd1 in TLR2/TLR4 agonists treated mouse macrophages compared with untreated control cells (Fig. 3F). These results indicate that Hrd1 is the pivotal E3 ubiquitin ligases responsible for the degradation of DDX5 during bacterial infection. Furthermore, during the treatment with inflammatory agonists (LPS, FSL-1, and Pam3CSK4), the K48-linked ubiquitination of DDX5 was enhanced in Hrd1-overexpressing MEFs but remained unchanged in Hrd1 mutant

(C329S)-overexpressing MEFs (Fig. 3G). Coincidently, the K48-linked ubiquitination of DDX5 was inhibited in Hrd1-silenced mouse macrophages or MEFs (Appendix Fig. S4D and Fig. 3H). Next, we found the recruitment of DDX5 to Hrd1 was significantly blocked in MEF cells with MyD88 knocked down by siRNA, while no such event was observed in TRIF-knocked down MEFs, upon stimulation by LPS, FSL-1 or Pam3CSK4 (Appendix Fig. S5A,B). This indicates that DDX5 was recruited to Hrd1 in an MyD88-dependent manner during bacterial infection. Overall, these findings demonstrate that bacterial infection or TLR activation induce DDX5 degradation via the ubiquitin-proteasome pathway and that Hrd1 is the bona fide E3 ligase for DDX5.

## DDX5 interacts with METTL14 to stabilize the DDX5-METTL3-METTL14 complex

To determine the precise role of DDX5 during inflammation, we investigated the binding partners of DDX5 upon bacterial infection. Previous studies showed DDX5 interacted with METTL3 and this association could affect the interaction between METTL3 and METTL14, which play essential roles in the m6A writer complex (Wei et al, 2021; Xu et al, 2021b; Zhao et al, 2022). However, it remained unclear how DDX5 modulated inflammation during bacterial infection. To test if DDX5 is involved in the formation of METTL3-METTL14 complex, we co-transfected 293T cells with plasmids encoding HA-tagged METTL14 and Flag-tagged DDX5 and observed interaction of METTL14 and DDX5 by co-immunoprecipitation (co-IP) analysis (Fig. EV2A). In addition, confocal microscopy was applied to assess the intracellular localization of ectopically expressed DDX5 and METTL14 and both proteins were found primarily in the nucleus of cells and co-localized (Fig. EV2B). Similar results were observed in mouse macrophages (Mø) and MEFs (Fig. 4A,B). Thus, we have shown the association of DDX5 and METTL14 by immunoaffinity purification and confirmed their intracellular colocalization using confocal microscopy.

Previous studies have demonstrated an interaction between DDX5 and METTL3, as well as an interaction between METTL3 and METTL14 to form a m6A writer complex upon virus infection (Zhao et al, 2018; Huang et al, 2019; Xu et al, 2021b). We therefore aimed to evaluate the possibility of DDX5 forming a complex with METTL3 and METTL14 during bacterial infection. Human 293T cells were co-transfected with plasmids encoding for Myc-METTL3, HA-METTL14 and Flag-DDX5 or the relative control plasmids and their interactions were assessed by co-IP analysis. Interestingly, the interaction between Myc-METTL3 and HA-METTL14, which was previously reported, was enhanced dramatically as the amount of Flag-DDX5 increased (Fig. EV2C). Similarly, the presence of METTL3 further strengthened the

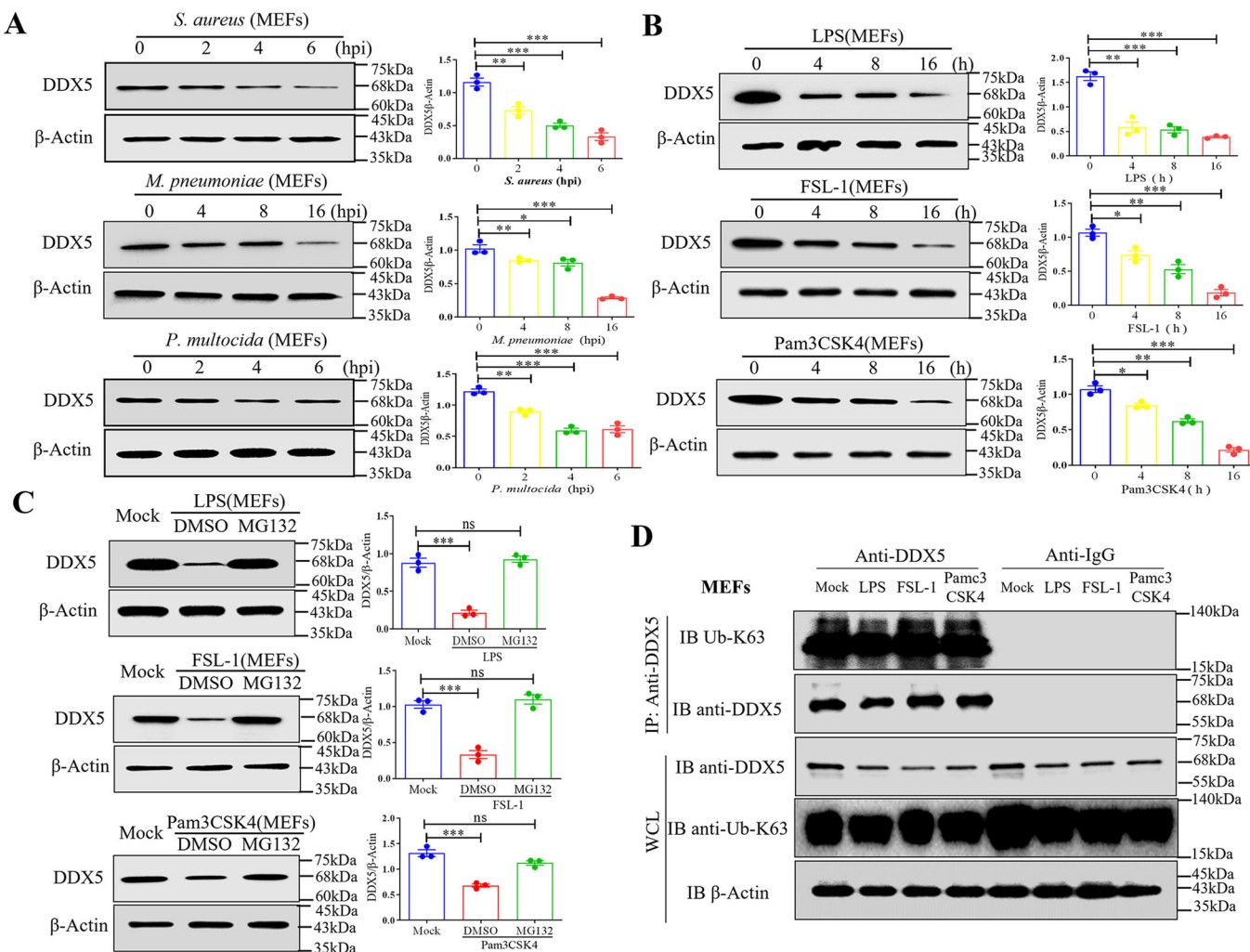

Figure 2. Bacterial infection led to DDX5 depression in K48-linked ubiquitin modification.

(A) DDX5 expression in MEFs infected with *S. aureus* for 0, 2, 4, and 6 h; *M. pneumoniae* for 0, 4, 8, and 16 h; *P. multocida* for 0, 2, 4, and 6 h. β-Actin as a reference control.
(B) DDX5 expression in MEFs treated with LPS, FSL-1, and Pam3CSK4 for 0, 4, 8, and 16 h. (C) DDX5 level in MEFs treated with the MG132 proteasome inhibitor, as well as LPS, FSL-1, and Pam3CSK4 for 8 h; DMSO served as the control. (D) Degradation of DDX5 in MEFs after treatment of TLR2/TLR4 agonists was not through K63 linked ubiquitin-proteasome pathway. The expression of DDX5 was quantified by the band intensity of DDX5/β-Actin in the western blot; the band intensity was measured by Image J software. Data information: In (A–C), all data are represented as the mean ± SEM of three biologically independent samples. "ns" indicates no significant difference, *p < 0.05, **p < 0.01, and ***p < 0.001 (Student's t test). Source data are available online for this figure.

interaction between DDX5 and METTL14 (Fig. EV2D). Furthermore, the association of DDX5 and METTL3 was substantially improved when METTL14 was co-transfected with the prior two (Fig. EV2E). As ectopic expression of these proteins may introduce artificial effect, we examined the interaction of the three proteins expressed endogenously in MEFs by transfecting only one at a time. The interaction between endogenous METTL14 and METTL3 was significantly enhanced when DDX5 was co-transfected in MEFs (Fig. 4C), but was significantly weakened when DDX5 was depressed by siRNA (Fig. 4D). The interaction between endogenous DDX5 and METTL14 was intensified when METTL3 was introduced (Fig. 4E), but was mitigated in METTL3 knockdown MEFs (Fig. 4F). In addition, the association between DDX5 and METTL3 was further reinforced in cells transfected with METTL14 (Fig. 4G) but significantly weakened in METTL14 knockdown

MEFs (Fig. 4H). These results suggested the presence of one of the three proteins served to strengthen the interactions between the other two. Furthermore, the association was not cell line specific as we were able to observe the co-localization of endogenous DDX5, METTL3, and METTL14 in both MEFs or mouse macrophages (Appendix Fig. S6). All these results demonstrated that DDX5 interacts with METTL14 to stabilize the DDX5-METTL3-METTL14 complex. Moreover, we found that the addition of TLR2/4 agonists, which led to the degradation of DDX5, significantly decreased the interaction between METTL3 and METTL14, which led to the impaired stability of the DDX5-METTL3-METTL14 complex in MEFs (Appendix Fig. S7A–C). These results indicate that the stability of the DDX5-METTL3-METTL14 complex is compromised during inflammatory activation.

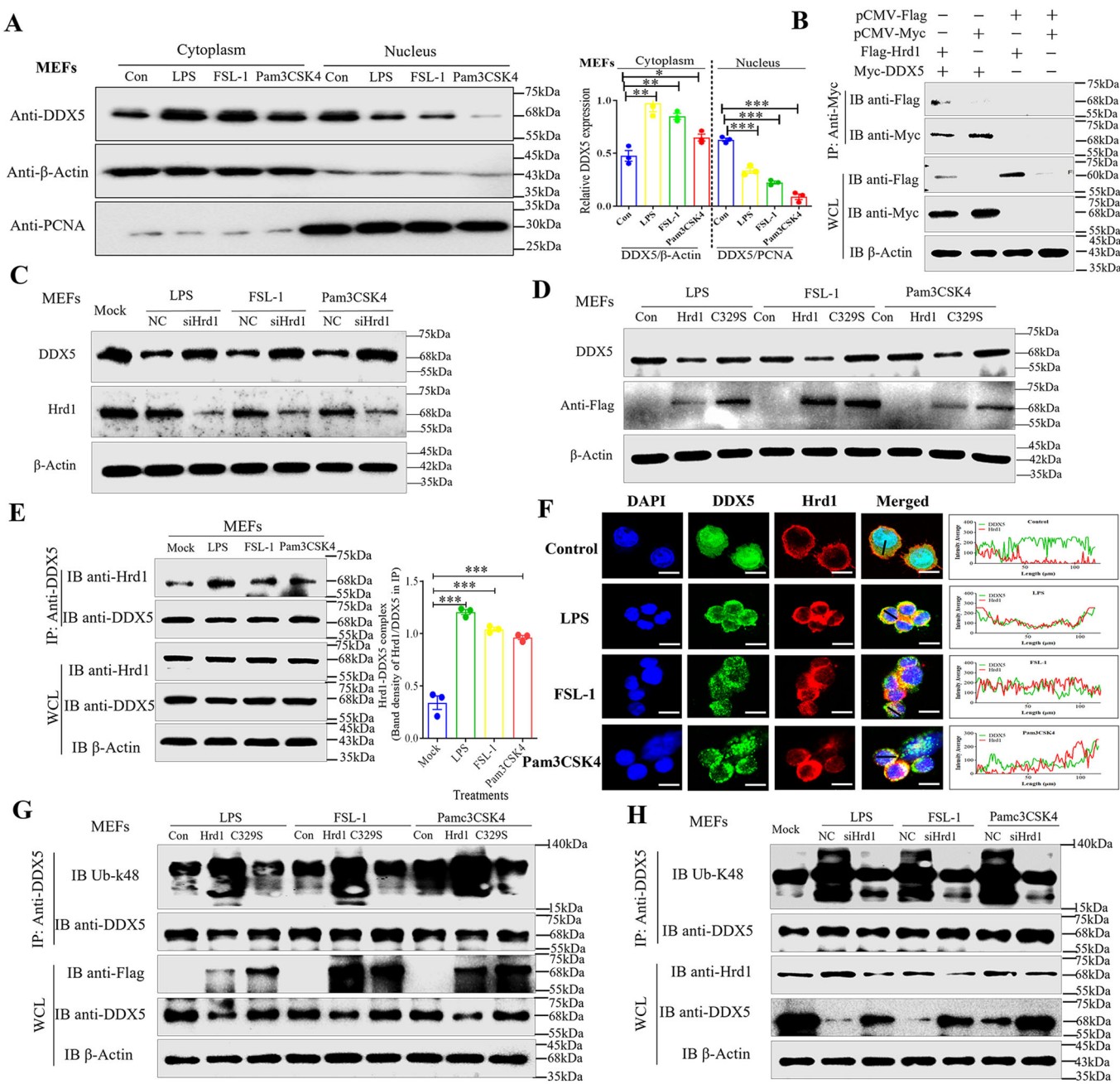

## DDX5 blocks inflammatory responses by regulating m6A modification of TLR2/4 transcripts

To assess the role of DDX5 in METTL3-METTL14-mediated m6A modification of mRNAs during the bacterial inflammatory response, we tested m6A modification in DDX5-deficient and DDX5-overexpressing cells. We observed that the m6A modification of mRNA in total RNA was higher in DDX5-overexpressing MEFs, but lower in MEFs in which the DDX5 was depressed by siRNA compared to that of the control cells (Fig. 5A). This result was also confirmed by an alternative approach showing enhanced m6A levels in cells with overexpressed DDX5 than the control cells and the cells in which DDX5 was suppressed (Fig. 5B). Next, to

identify the cellular target of the DDX5-METTL3-METTL14 m6A writer complex, we performed a high-throughput m6A sequencing using MEFs treated with LPS, FSL-1, and Pam3CSK4. All treated MEFs were analyzed with respect to a characteristic "GGACU" motif (Fig. 5C). Based on the MeRIP-seq and RNA-seq data (logFC ≥ 2.5), we found that DDX5 affected a variety of transcripts in both m6A modification and transcription (Appendix Fig. S8A–C, $p < 0.01$), including those of TLR2 and TLR4. We found the TLR4 transcripts exhibited an obvious peak denoting m6A modification, which was much lower in DDX5-knockdown MEFs than that of the negative controls, a similar m6A peak was detected for the TLR2 transcripts, which was also blunted when siDDX5 was transfected in the cells but not the siNC controls (Fig. 5D). The

**Figure 3. DDX5 is degraded by Hrd1 during bacterial induced inflammation.**

(A) Redistribution of DDX5 after treatment of TLR2/TLR4 agonists. MEFs were treated with LPS, FSL-1, and Pam3CSK4 respectively, harvested to extract the nuclear and cytoplasmic protein for western blot assay to detect the expression of DDX5.β-Actin acted as a cytoplasmic reference control, and PCNA was a nuclear reference control. The expression of cytoplasmic DDX5 was quantified by the relative intensity of DDX5/β-Actin, and the expression of nuclear DDX5 was quantified by the relative intensity of DDX5/PCNA. The band intensity was measured by Image J software. (B) Interaction between Flag-Hrd1 and Myc-DDX5 in 293T cells detected by Co-IP. (C) DDX5 level in Hrd1 knockdown MEFs after treatment with LPS, FSL-1, and Pam3CSK4; NC served as the siRNA negative control. (D) DDX5 level in Flag-Hrd1- or Flag-Hrd1mutant (C329S)-expressing MEFs after treatment with LPS, FSL-1, and Pam3CSK4; Con served as the vector control. (E) Interaction between Hrd1 and DDX5 in MEFs after treatment with LPS, FSL-1, and Pam3CSK4 by Co-IP with an anti-DDX5 antibody. The interaction of Hrd1 and DDX5 was quantified by the band density of Hrd1/DDX5 in the IP system; the band density was measured by Image J software. (F) The co-localization of DDX5 and Hrd1 in mouse macrophages (Mø) after treatment of TLR2/TLR4 agonists. The mouse macrophages (Mø) were treated with TLR2/TLR4 agonists for 4 h, then fixed with 4% paraformaldehyde and stained with anti-DDX5 and anti-Hrd1before confocal microscopy. Right panels show the pixel intensity of red (Hrd1) and green (DDX5) from the black line in the merged image. Scale bars: 10 μm. (G) The K48 linked ubiquitin modification of DDX5 in Flag-Hrd1- or Flag-Hrd1 mutant (C329S)-expressing MEFs after treatment with LPS, FSL-1, and Pam3CSK4; Con served as the vector control. (H) The K48-linked ubiquitin modification of DDX5 in Hrd1 knockdown MEFs after treatment with LPS, FSL-1, and Pam3CSK4; NC served as the siRNA negative control. Data information: In (A, E), all data are represented as the mean ± SEM of three biologically independent samples. "ns" indicates no significant difference, $*p < 0.05$, $**p < 0.01$ and $***p < 0.001$ (Student's $t$ test). Source data are available online for this figure.

m6A pattern was located in the PR_intron of TLR4 transcript when MEFs were treated with LPS, which was otherwise located in the CDS of TLR2 transcript when MEFs were treated with FSL-1 and Pam3CSK4 (Fig. 5D). In order to confirm the m6A modification on TLR2/4 transcripts, we constructed mutants containing mutations in the m6A site of TLR2 and TLR4 transcripts by replacing the Adenine (A) with Uracil (U), Guanine (G) or Cytosine (C), and examined their binding capability with METTL3 (Appendix Fig. S8D). In Appendix Fig. S8E, in comparison to WT m6A *tlr4* with WT m6A *tlr2*(m6A *tlr2-1* and m6A *tlr2-2*), the m6A *tlr4* mutant (m6A *tlr4*-M) and m6A *tlr2* mutants (m6A *tlr2-1*-M and m6A *tlr2-2*-M) completely lost the binding ability of METTL3 in MEFs upon treated with LPS, FSL-1 or Pam3CSK4. These results demonstrated the sites identified were indeed m6A modification sites and were recognized and targeted by METTL3. Importantly, other than the m6A sites on TLR2/4 transcripts, we did not find any other TLR transcripts whose m6A modification was influenced by DDX5 after treatment of TLR2 and TLR4 agonists (Appendix Fig. S8A–C, $p < 0.01$). This indicates that DDX5 specifically controlled m6A modification of TLR2/4 transcripts. To further confirm this observation, we set GAPDH and TAK1 as negative controls and calculated the relative m6A levels of TLR2 and TLR4 by normalizing their m6A levels to that of GAPDH and TAK1. We monitored m6A-enriched TLR2/4 transcripts in MEFs with overexpressed or decreased levels of DDX5 or METTL3 by RNA immunoprecipitation quantitative PCR (RIP-qPCR). Compared with the negative control (NC), the level of m6A on TLR4 transcripts upon LPS treatment decreased significantly when DDX5 or METTL3 was knockdown (Fig. 5E). Similar results were observed when cells were treated with FSL-1 or Pam3CSK4 instead of LPS (Fig. 5E). In comparison, levels of m6A on TLR4 transcripts upon LPS treatment displayed a significant enhancement in cells overexpressing DDX5 or METTL3; a similar trend was recorded on TLR2 transcripts when FSL-1 or Pam3CSK4 were used to stimulate inflammation (Fig. 5F). These results demonstrated that DDX5 and its binding partner METTL3 specifically controlled m6A modification of TLR2/4 transcripts when TLR2 and TLR4 were activated by agonists. Previous studies showed that mRNA m6A modification mediated by METTL3 necessitated direct binding of transcripts to METTL3 protein (Xiong et al, 2022). To examine if the writer complex comprising of DDX5 also directly binds to TLR2/4 transcripts, we employed individual nucleotide-resolution cross-linking and immunoprecipitation quantitative PCR (iCLIP-qPCR)

to determine the binding of transcripts to METTL3 following the treatment with TLR2/4 agonists. We found that while m6A levels on GAPDH and TAK1 transcripts were unaffected by the amount of DDX5 present in the cells, depression of DDX5 significantly decreased the binding of TLR4 and TLR2 transcripts with METTL3 (Fig. 5G) and overexpressing DDX5 enhanced the RNA-protein association (Fig. 5H). These results demonstrated that DDX5 promoted the m6A modification on TLR2/4 transcripts by enhancing the binding of target transcripts to the writer complex. To determine if DDX5-promoted m6A modification affected the levels of TLR4 and TLR2 transcripts, we determined the abundance of targeted transcripts by qRT-PCR in cells with depressed DDX5 or METTL3. As shown in Fig. 5I, the mRNA level of TLR4 was significantly higher in DDX5 or METTL3 knockdown MEFs treated with LPS but significantly lower in MEFs overexpressing either DDX5 or METTL3 upon LPS treatment (Fig. 5J). An analogous trend was observed upon treatment with FSL-1 or Pam3CSK4 (Fig. 5I,J). These results suggest that DDX5-promoted m6A negatively regulates the levels of TLR4 and TLR2 transcripts. We next examined the protein levels of TLR4 and TLR2 and discovered that consistent with the results with mRNA abundance, the amounts of METTL3 and DDX5 were negatively correlated with the protein levels of TLR4 and TLR2 (Fig. EV3A,B). An alternative approach using confocal microscopy was employed to examine the effect of DDX5 on TLR2 and TLR4 expression and the results confirm previous observations, showing that ectopic expression of DDX5 leads to decreased levels of TLR2 and TLR4 (Fig. EV3C).

Pathogenic bacteria are mainly recognized by TLR2/4, thus triggering the TLR2/4-mediated inflammatory responses (Adelaja and Hoffmann, 2019; Sabroe et al, 2008). Next, we sought to determine if DDX5 affects the expression of TLR2, TLR4 *ex vivo*. Primary BMDM from DDX5$^{+/+}$ and DDX5$^{+/-}$ mice were collected and treated with LPS, FSL-1, or Pam3CSK4 for 0, 4, and 6 h, and the protein levels of TLR4, TLR2 and DDX5 were monitored by Western blotting analysis. Compared with DDX5$^{+/+}$ BMDM, the expression level of TLR4 was higher in DDX5$^{+/-}$ BMDM following LPS treatment (Appendix Fig. S9A). Similarly, the expression level of TLR2 was also higher in DDX5$^{+/-}$ BMDM following FSL-1 or Pam3CSK4 treatment than that in DDX5$^{+/+}$ cells (Appendix Fig. S9A). Furthermore, we measured the synthesis of IL-6 and TNF-α in both DDX5$^{+/+}$ and DDX5$^{+/-}$ BMDM upon treatment of TLR agonists including LPS, FSL-1 or Pam3CSK4 for 0, 4, and 6 h with ELISA. Compared with DDX5$^{+/+}$ BMDM, the production of IL-6 (Fig. EV3D) or TNF-α (Fig. EV3E) were

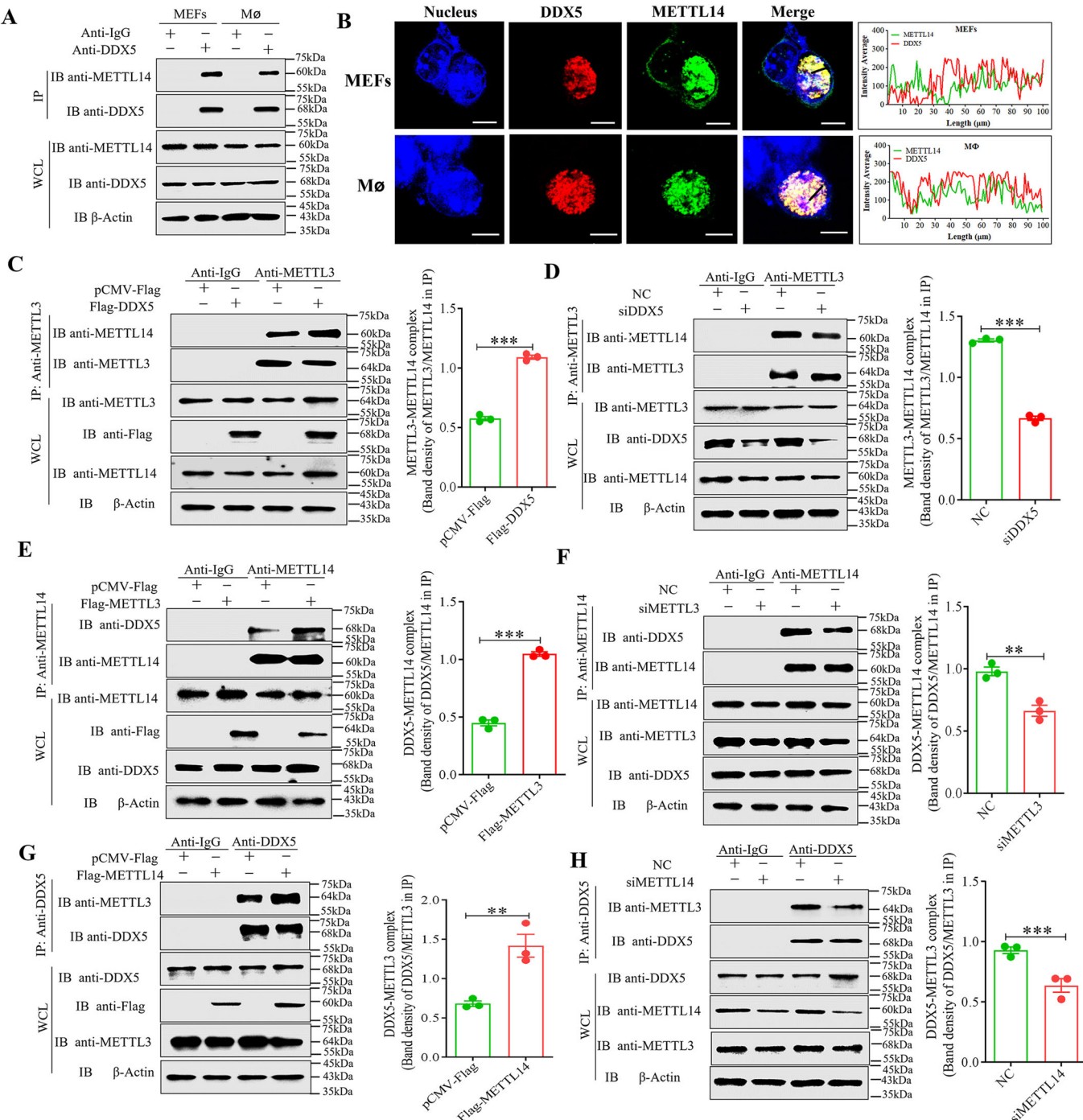

**Figure 4. DDX5 interacts with METTL14 to stabilize the DDX5-METTL3-METTL14 complex.**

(A) Interaction between DDX5 and METTL14 in MEFs and mouse macrophages (Mø) detected by Co-IP. (B) Co-localization of METTL14 and DDX5 in MEFs or mouse macrophages (Mø). The MEFs or mouse macrophages (Mø) were seeded on coverslips in 12-well plates and cultured for 18–24 h, then fixed with 4% paraformaldehyde and stained with anti-DDX5 and anti-METTL14 before confocal microscopy. Right panels show the pixel intensity of red (DDX5) and green (METTL14) from the black line in the merged image. Scale bars: 7.5 μm. (C, D) Interaction between METTL3 and METTL14 in control and DDX5-overexpressing MEFs (C) or control (NC) and DDX5 knockdown (siDDX5) MEFs (D) by Co-IP with an anti-METTL3 antibody. (E, F) Interaction between METTL14 and DDX5 in control and METTL3-overexpressing MEFs (E) or NC and METTL3 knockdown (siMETTL3) MEFs (F) by Co-IP with an anti-METTL14 antibody. (G, H) Interaction between METTL3 and DDX5 in control and METTL14-overexpressing MEFs (G) or NC and siMETTL14 MEFs (H) by Co-IP with an anti-DDX5 antibody. Data information: In (C–H), all data are represented as the mean ± SEM of three biologically independent samples. **$p < 0.01$ and ***$p < 0.001$ (Student's $t$ test). Source data are available online for this figure.

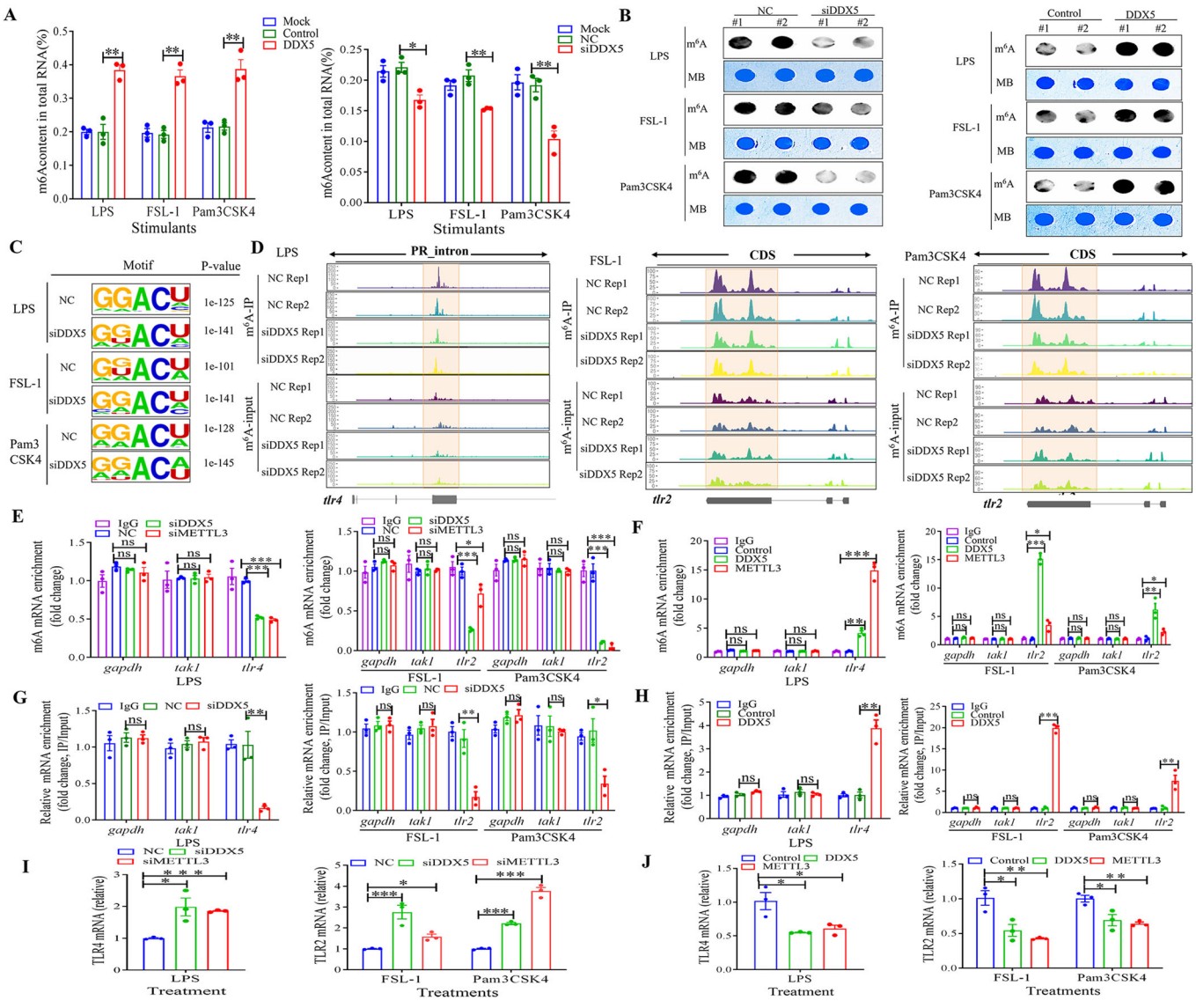

**Figure 5. The m6A modification of TLR2/4 mRNA is regulated by DDX5 during bacterial inflammatory response.**

(A) m6A content of total RNA in control and DDX5-overexpressing MEFs or NC and siDDX5 MEFs treated with LPS, FSL-1, and Pam3CSK4 measured by nucleotide array (n = 3). (B) m6A content of total RNA in NC and siDDX5 MEFs or control and DDX5-overexpressing MEFs treated with LPS, FSL-1, and Pam3CSK4 measured by dot-blot assays. Methylene blue (MB) was used as a reference control. (C) Sequence motifs identified within m6A peaks in MEFs from two biological replicates with the lowest p value. (D) Piled reads of TLR4/TLR2 transcripts in DDX5 knockdown (siDDX5) MEFs treated with LPS, FSL-1, or Pam3CSK4 and identified by m6A sequencing. (E, F) Enrichment with m6A-tlr4/m6A-tlr2 mRNA in NC, siDDX5, and siMETTL3 MEFs (E) or in control, DDX5-overexpressing, and METTL3-overexpressing MEFs (F) treated with LPS, FSL-1, or Pam3CSK4 and determined by RIP-qPCR (n = 3). (G, H) Binding of TLR4/TLR2 mRNA to METTL3 in NC and siDDX5 MEFs (G) in control and DDX5-overexpressing MEFs (H) treated with LPS, FSL-1, or Pam3CSK4 as determined by iCLIP-qPCR.The GAPDH mRNA (gapdh) served as the negative control, and TAK1 mRNA (tak1) was the no-m6A modified mRNA control. (I, J) Relative TLR4/TLR2 mRNA content in NC, siDDX5, and siMETTL3 MEFs (I) or in control, DDX5-overexpressing, and METTL3-overexpressing MEFs (J) treated with LPS, FSL-1, or Pam3CSK4 and measured by qPCR. Data information: In (A, E–J), all data are represented as the mean ± SEM of three biologically independent samples."ns" indicates no significant difference, *p < 0.05, **p < 0.01, and ***p < 0.001 (Student's t-test). Source data are available online for this figure.

consistently and significantly greater in DDX5⁺/⁻ BMDM, regardless of which TLR agonist used; these results indicated that primary DDX5 in BMDM cells suppressed the expression of TLR2/4. Next, we decided to examine the effect of DDX5 on the expression of TLR2 and TLR4 in vivo. DDX5⁺/⁻ and DDX5⁺/⁺ mice were infected with *M. pneumoniae*, *S. aureus*, or *P. multocida*, and lung tissue was collected and subjected to western blotting analysis for the expression of DDX5, TLR2/4 or immunohistochemical analysis. Compared with DDX5⁺/⁺

mice, DDX5 was expressed at a much lower level in the lungs of DDX5⁺/⁻ mice, whereas the opposite was observed for TLR4 during 0–24 h post-infection with *P. multocida* (Appendix Fig. S9B). A similar discrepancy in TLR2 expression was also observed in the lungs of DDX5⁺/⁻ mice comparing to DDX5⁺/⁺ mice during 0–36 h post-infection with *M. pneumoniae* and during 0–24 h post-infection with *S. aureus* (Appendix Fig. S9B). Furthermore, the increased expression of TLR4 in the lungs of DDX5⁺/⁻ mice upon infection with *P.*

multocida and of TLR2 upon infection with *S. aureus* and *M. pneumoniae* was confirmed by immunohistochemistry staining (Appendix Fig. S9C). Consistently, the strong production of inflammatory cytokines (IL-6/TNF-α) was significantly higher in the serum and lungs of DDX5[+/-] mice (Fig. 1A–D), as well as inflammatory response in DDX5[+/-] mice (Fig. 1E) than that in DDX5[+/+] mice. All these results indicate that DDX5 suppresses the expression of TLR2 and TLR4 as well as their downstream NF-κB-mediated inflammatory signaling pathway in vivo.

## DDX5 promotes RNA decay of m6A-modified TLR2/TLR4 transcripts in a YTHDF2-dependent manner

The fate of m6A-modified mRNAs is determined by cytoplasmic reader proteins, which dictate mRNA stability, degradation and translation (Adelaja and Hoffmann, 2019). Therefore, we evaluated the stability of TLR2/4 transcripts in DDX5-overexpressing or DDX5-deficient MEFs treated with TLR2/4 agonists. RNA decay assays revealed that TLR4 mRNA decayed more rapidly in DDX5-overexpressing MEFs treated with LPS than in controls, the same trend was observed for TLR2 mRNAs in DDX5-overexpressing MEFs subjected to FSL-1 or Pam3CSK4 treatment (Fig. 6A). Conversely, a significantly slower mRNA decay was detected for TLR4 transcripts in DDX5-deficient MEFs treated with LPS and for TLR2 transcripts in DDX5-deficient MEFs treated with FSL-1 or Pam3CSK4 (Fig. 6B). These findings indicate that DDX5 promotes the decay of TLR2/4 mRNA during TLR activation by agonists.

Previous studies identified YTHDF2 as the primary reader protein responsible for the degradation of cytoplasmic m6A-modified mRNA (Hazra et al, 2019). To explore if YTHDF2 is involved in DDX5-mediated degradation of TLR2/4 transcripts, we detected the binding of YTHDF2 to TLR2/4 mRNA in DDX5-overexpressing or DDX5-deficient MEFs stimulated with LPS, FSL-1, or Pam3CSK4 via a RIP qRT-PCR assay described above. The association between YTHDF2 and TLR4 transcripts was significantly enhanced when DDX5 was overexpressed concomitantly in MEFs following LPS treatment, as well as TLR2 transcripts treated with FSL-1 or Pam3CSK4. However, no discernible difference in the binding of GAPDH transcripts to YTHDF2 was recorded between DDX5-overexpressing and DDX5-deficient cells, the binding of YTHDF2 to TLR2/TLR4 transcripts was also detected in DDX5-deficient MEFs treated with LPS, FSL-1, or Pam3CSK4, respectively, compared with the GAPDH control (*gapdh*), the binding of TLR4 transcripts to YTHDF2 was significantly decreased in LPS-treated DDX5-deficient MEFs, and the binding of TLR2 transcripts to YTHDF2 was also significantly decreased in FSL-1/Pam3CSK4-treated DDX5-deficient MEFs (Fig. 6C). These results showed that DDX5 could regulate the binding of YTHDF2 to TLR2/4 transcripts to affect the stability of TLR2/4 mRNAs. Moreover, the level of TLR4 mRNA was increased when YTHDF2 was silenced by siRNA in MEFs upon LPS treatment, and the same was observed for TLR2 mRNA upon FSL-1 or Pam3CSK4 treatment (Fig. 6D). TLR4 and TLR2 were also expressed at higher levels in YTHDF2-silenced MEFs than in siRNA-treated controls following LPS treatment (Appendix Fig. S10A) or FSL-1 and Pam3CSK4 (Appendix Fig. S10B,C), respectively. Akin to MEFs, the expression of TLR4 also increased in YTHDF2-silenced mouse macrophages under LPS treatment, while the expression of TLR2 improved in YTHDF2-silenced mouse macrophages under FSL-1

or Pam3CSK4 treatment (Fig. EV4A). All these results showed that the knockdown of YTHDF2 could effectively improve the expression of TLR2/4 in MEFs or mouse macrophages. Finally, an assessment of IL-6 levels revealed that cytokine production was significantly higher in YTHDF2-silenced MEFs (Fig. 6E) or YTHDF2-silenced mouse macrophages (Fig. EV4B) upon treatment with LPS, FSL-1, and Pam3CSK4. Also, silencing of YTHDF2 effectively blocked the inhibition of IL-6 production in DDX5/METTL3-overexpressing MEFs under the treatment of LPS or Pam3CSK4 (Appendix Fig. S11). Taken together, these results confirm that DDX5 promotes the degradation of TLR2/4 transcripts in a YTHDF2-dependent manner.

## Depletion of DDX5/METTL3/METTL14 m6A machinery leads to elevated TLR2/4/NF-κB-mediated inflammatory responses during bacterial infection in vivo

Previous studies have shown that METTL3 is the binding protein of DDX5 in mRNA N6-methyladenosine modification (Zhao et al, 2018; Xu et al, 2021b; Zhao et al, 2022). In this study, we found that DDX5 regulates the stability and enzymatic activity of a methyltransferase complex comprised of DDX5, METTL3 and METTL14. However, whether the same DDX5/METTL3-mediated m6A machinery also regulates TLR2/4/NF-κB-mediated inflammatory responses during bacterial infection was unclear. Given the fact that METTL3-mutant mice homozygous for a knockout allele exhibit embryonic lethality between E3.5 and E8.5 with a deficiency in adopting the epiblast egg cylinder, we decided to generate a line of mice with a conditional knockout of METTL3 (designated *Mettl3* cKO) to assess the effect of METTL3 on TLR activation and NF-κB activation. Moreover, *Mettl3* cKO progeny have similar body weight to that of *Mettl3* wild type (WT) littermates, and can reach adulthood (Fig. EV5A); moreover, there was no difference in the organ coefficient (organ-body rate) between *Mettl3* cKO mice and wild type littermates (Fig. EV5B). Therefore, we employed *Mettl3* cKO mice to verify the role of DDX5/METTL3-mediated m6A machinery during bacterial infection in vivo. First, primary BMDM from *Mettl3* cKO and *Mettl3* WT mice were collected and maintained in cell culture prior to infection with *M. pneumoniae*, *S. aureus* and *P. multocida*. PBS was used as the control treatment. Infected cells were then lysed and subjected to SDS-PAGE and Western blotting analysis to detect TLR4, TLR2, phosphorylation of p65 and METTL3. Compared with primary *Mettl3* WT BMDM, METTL3 expression was nearly undetected in primary *Mettl3* cKO BMDM, whereas the protein level of TLR4 and phosphorylation of p65 were greater in BMDM collected from *Mettl3* cKO mice than in *Mettl3* WT mice when stimulated with *P. multocida* (Appendix Fig. S12A). A similar increase in TLR2 and p-p65 expression was detected in primary *Mettl3* cKO BMDM during 0 ~ 36 h post-infection with *M. pneumoniae* (Appendix Fig. S12B) and during 0–24 h post-infection with *S. aureus* (Appendix Fig. S12C). All these results indicated that lack of METTL3 induces an activation of the TLR2/4/NF-κB-mediated signaling pathway in primary BMDM. Consistently, we observed significantly higher IL-6 levels in primary *Mettl3* cKO BMDM compared to *Mettl3* WT BMDM following infection with *P. multocida* and *S. aureus* for 0–12 h, and *M. pneumoniae* for 0–24 h (Appendix Fig. S12D), as well as TNF-α (Appendix Fig. S12E). Moreover, the increased expression of TLR4 in the lungs of *Mettl3* cKO mice upon infection with *P. multocida*

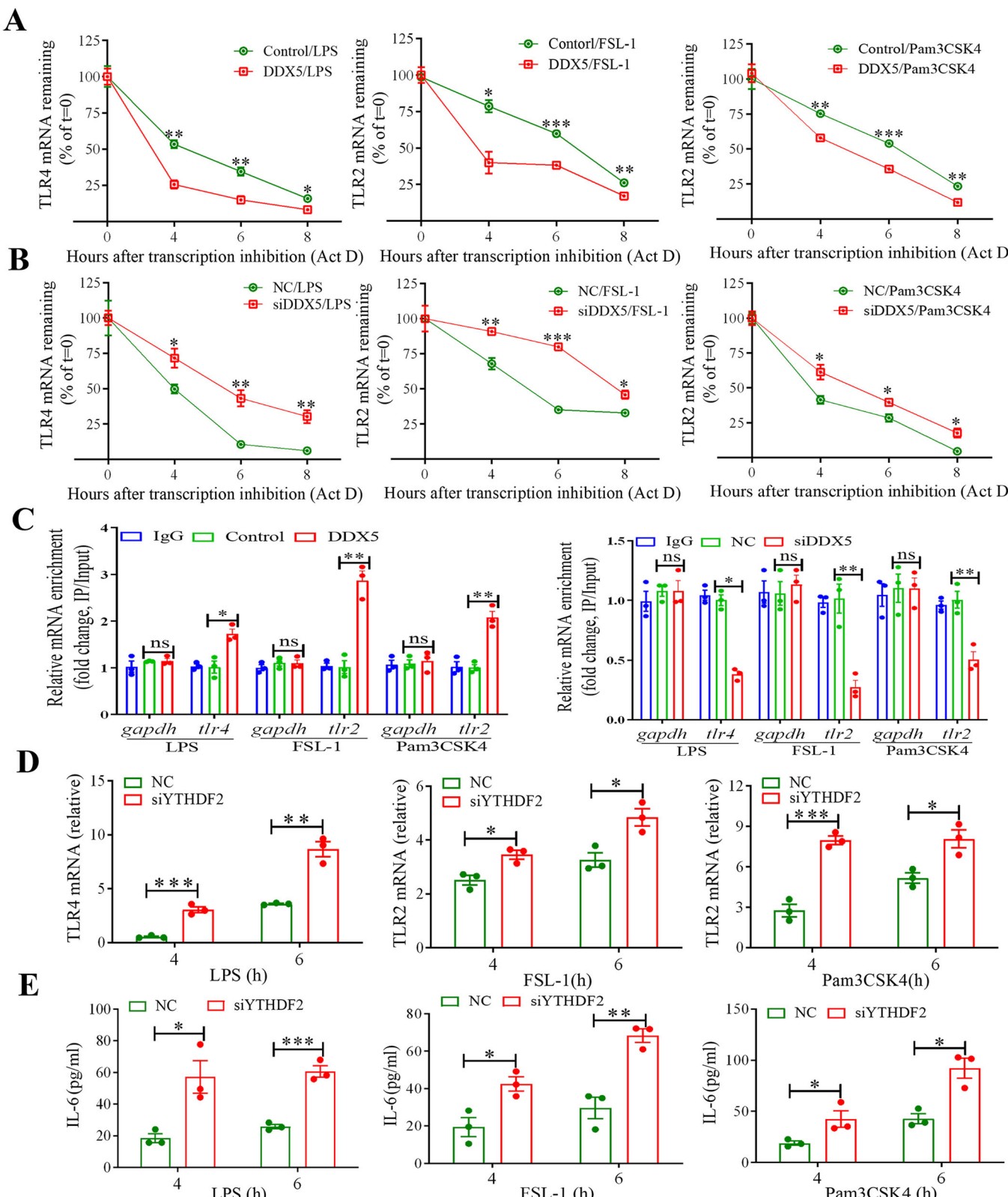

◄ **Figure 6.  DDX5 suppresses the production of IL-6 by promoting RNA decay of TLR2/4 transcripts via YTHDF2.**

(A) TLR4/TLR2 mRNA decay assay in control and DDX5-overexpressing MEFs treated with LPS, FSL-1 or Pam3CSK4. (B) TLR4/TLR2 mRNA decay assay in control (NC) and DDX5 knockdown (siDDX5) MEFs treated with LPS, FSL-1 or Pam3CSK4. (C) Binding of TLR4/TLR2 mRNA to YTHDF2 in control and DDX5-overexpressing MEFs or NC and siDDX5 MEFs treated with LPS, FSL-1 or Pam3CSK4 and assessed by iCLIP-qPCR. (D) Relative TLR4/TLR2 mRNA content in NC and YTHDF2 knockdown (siYTHDF2) MEFs treated with LPS, FSL-1 or Pam3CSK4 and measured by qPCR. (E) IL-6 production in NC and siYTHDF2 MEFs treated with LPS, FSL-1, and Pam3CSK4. Data information: In (A–E), all data are represented as the mean ± SEM of three biologically independent samples. "ns" indicates no significant difference, *$p < 0.05$,**$p < 0.01$, and ***$p < 0.001$ (Student's $t$ test). Source data are available online for this figure.

and of TLR2 upon infection with *S. aureus* and *M. pneumoniae* were confirmed by immunohistochemistry staining (Fig. 7A). And the levels of IL-6 and TNF-α were significantly higher in serum (Fig. 7B,C) and lungs of *Mettl3* cKO mice than those of *Mettl3* WT mice upon infection with *P. multocida*, *S. aureus*, and *M. pneumoniae* (Fig. 7D,E). Finally, The stronger inflammatory response in the lungs of *Mettl3* cKO mice relative to *Mettl3* WT mice infected with pathogenic bacteria was confirmed by hematoxylin and eosin tissue staining (Fig. 7F), while a significant decrease of bacterial load in the lung of *Mettl3* cKO mice was observed (Appendix Fig. S13). All these results suggest that DDX5/METTL3/METTL14-mediated m6A modification suppresses the TLR2/4/NF-κB-mediated inflammatory signaling pathway in vivo during bacterial infection.

## Discussion

Several RNA helicases, including DDX1, DDX21, DHX36, DDX58, DDX3, DHX9, DDX41, and DDX25, have been implicated in immune regulation, and nearly all of them participate in antiviral innate immunity through regulation of type I interferon response (Fuller-Pace, 2006; Baldaccini and Pfeffer, 2021; Tang and Zheng, 2021; Ullah et al, 2022). Instead, the role of DExD/H-box helicases in the inflammatory response to bacterial infection remains unclear, and no RNA helicase has been reported to participate in such a process. In this study, we demonstrated that DDX5 suppressed inflammatory responses by downregulating the levels of TLR2/4 transcripts via mRNA m6A modification. During bacterial infection, DDX5 was recruited to and degraded by an ER-localized E3 ligase Hrd1 via the ubiquitin-proteasome pathway. The loss of DDX5 disturbed the writer complex comprised of DDX5, METTL3 and METTL14 and prevented m6A modification of TLR2/4 mRNA, thereby resulting in increased expression of TLR2/4, the main PAMPs for bacterial-induced inflammation (Sabroe et al, 2008). The activation of TLR2 and TLR4 then induced the phosphorylation of p65 and the subsequent inflammatory response. This is not only the first report of a pivotal role played by RNA helicases in response to bacterial infection but also exhibited the role of DDX5 in modulating inflammation quiescence during homeostasis. Unlike DHX29 or DDX19A, which act as viral RNA sensors and activators of inflammation through the NF-κB or NLRP3 pathways (Li et al, 2015; Zhu et al, 2018), DDX5 regulates the m6A modification of TLR2/4 mRNA to suppress inflammatory responses pathway during bacterial infection. Though DDX5 was previously found to regulate the expression of DHX58 and NF-κB to impair antiviral innate immunity (Xu et al, 2021b); and mice with keratinocyte-specific deletion of DDX5 were more susceptible to cutaneous inflammation (Ni et al, 2022). Here, DDX5 was found to regulate inflammation caused by bacterial infection, indicating DDX5 may control diverse mechanisms in response to pathogenic infection.

Even though several studies have described the mechanism triggering inflammation during viral and bacterial infection (Sabroe et al, 2008; Meizlish et al, 2021), the switch between homeostasis and inflammation remains poorly explored. In this study, we measured the synthesis of IL-6 and TNF-α by MEFs and mouse macrophages, which are the major inflammatory cytokines (Jones and Jenkins, 2018). Using *P. multocida*, *S. aureus*, and *M. pneumoniae* as representative inducers of severe inflammation and pathological injury in humans and animals (Vujic et al, 2015; Peng et al, 2019; Bilyk et al, 2022), we observed robust IL-6 release in DDX5-silenced MEFs or mouse macrophages. Moreover, we verified the regulation of inflammation by DDX5 in vivo using knockout mice infected with pathogenic bacteria. Compared with wide type DDX5$^{+/+}$ mice, the inflammatory response in the lungs and serum of DDX5$^{+/-}$ mice were characterized by higher inflammatory cytokine levels and more severe tissue inflammation. All these results indicated that DDX5 was a crucial regulator of bacterial inflammation. Given the prominent role of TLR2/4 in recognizing bacterial PAMPs and activating inflammation through the NF-κB pathway (Adelaja and Hoffmann, 2019; Meizlish et al, 2021), the TLR2/4 agonists LPS, FSL-1, and Pam3CSK4 were used to explore the mechanism through which DDX5 activated the TLR2/4/NF-κB pathway upon bacterial infection. Because the expression of DDX5 decreased following treatment with TLR2/4 agonists in both MEFs and mouse macrophages, we hypothesized that post-translational modifications, such as phosphorylation, SUMOylation, or ubiquitination could be involved in down-regulating DDX5 upon bacterial infection (Clark et al, 2008; Li et al, 2021; Zheng et al, 2021). Indeed, we found DDX5 was translocated from the nucleus and targeted by an ER-localized E3 ligase Hrd1, which was the main E3 ligase responsible for degradation of nuclear proteins, such as USP15 and METTL14 (Lu et al, 2019; Wei et al, 2021). Furthermore, DDX5 underwent K48-linked ubiquitination, which then facilitated its degradation via the proteasome during bacterial infection.

RNA m6A methylation is involved in several aspects of inflammation regulation (Luo et al, 2021). The IGF2BP2 m6A reader stabilizes TSC1 and PPAR-γ to activate the inflammatory response (Wang et al, 2021), METTL14 promotes the m6A modification of FOXO1 mRNA to aggravate endothelial inflammation (Jian et al, 2020), and the YTHDF2 m6A reader suppresses pro-inflammatory pathways and sustains hematopoietic stem cell function (Mapperley et al, 2021). This evidence suggests that RNA m6A methylation may be a vital factor during the inflammatory process. Previously, DDX5 was found to target METTL3, thereby regulating the interaction between METTL3 and METTL14 (Xu et al, 2021b). Here, we found that DDX5 interacted not only with METTL3 but also with METTL14 to form a stable complex. Although a previous study showed that DDX5 worked as a mediator of the back-splicing reaction and as a co-factor of the

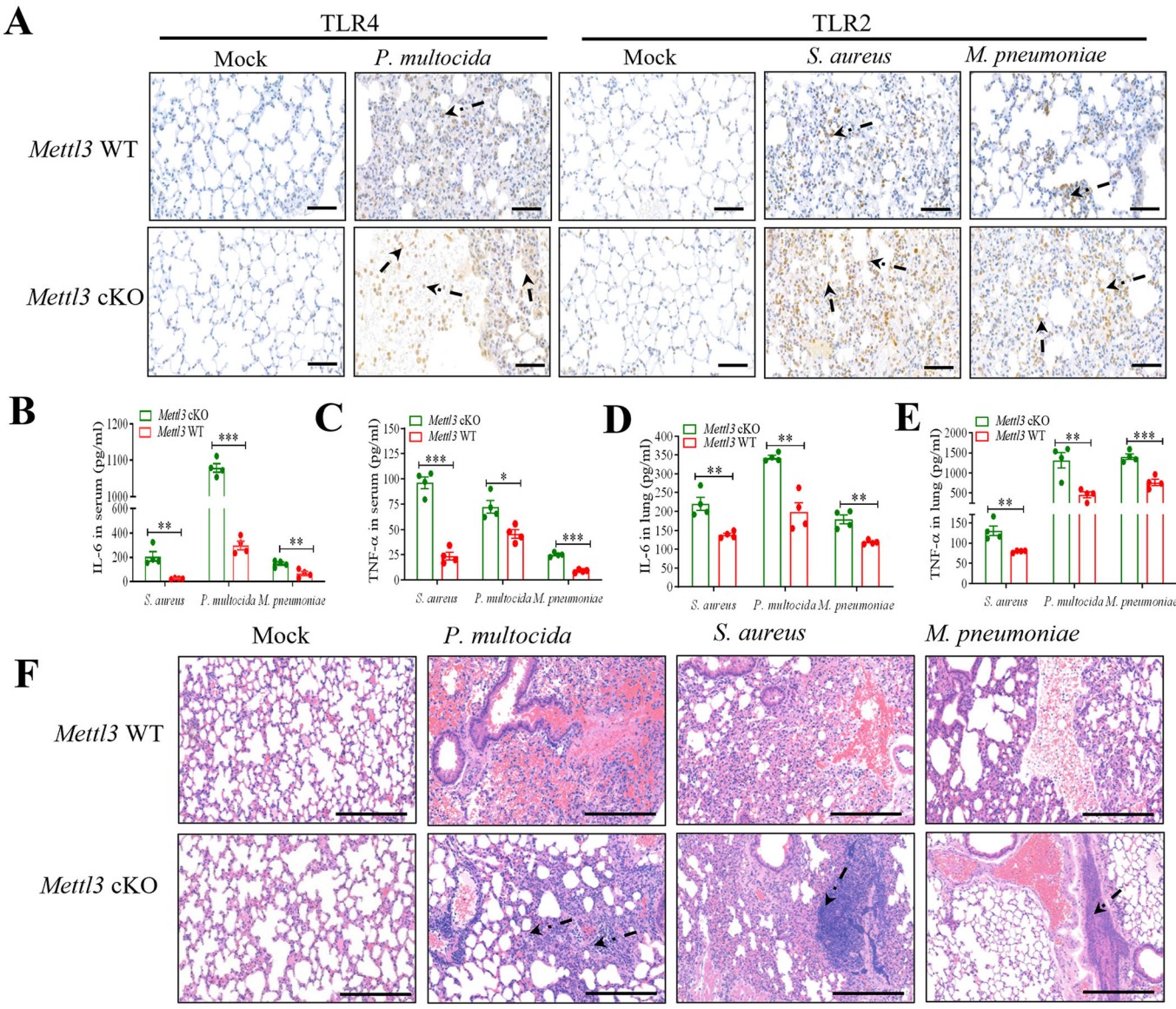

**Figure 7. DDX5-restrained expression of TLR4/TLR2 via METTL3-mediated m6A modification negatively regulates the bacterial inflammatory response in vivo.**

(A) Immunohistochemistry analysis of TLR4 in the lungs of *Mettl3* WT or *Mettl3* cKO mice infected with *P. multocida* for 12 h, and of TLR2 in the lungs of *Mettl3* WT or *Mettl3* cKO mice infected with *S. aureus* for 12 h and *M. pneumoniae* for 24 h. Scale bars, 50 μm. Black arrows indicated strong positive signals. (B, C) IL-6 (B) or TNF-α (C) production in serum of *Mettl3* WT mice or *Mettl3* cKO mice infected with *P. multocida*, *S. aureus*, or *M. pneumonia*, respectively (n = 4). (D, E) IL-6 (D) or TNF-α (E) production in the lungs of *Mettl3* WT mice or *Mettl3* cKO mice infected with *P. multocida*, *S. aureus*, or *M. pneumonia*, respectively (n = 4). (F) Hematoxylin and eosin staining of lung tissue from *Mettl3* WT or *Mettl3* cKO mice infected with *P. multocida* for 12 h, *S. aureus* for 12 h, and *M. pneumoniae* for 24 h. Black arrows indicate severe inflammatory pathology. Scale bars, 100 μm. Data information: In (B–E), all data are represented as the mean ± SEM of three biologically independent samples. *$p < 0.05$, **$p < 0.01$, and ***$p < 0.001$ (Student's *t* test). Source data are available online for this figure.

m6A regulatory network (Dattilo et al, 2023), our data provided the first piece of evidence showing that RNA helicases regulate mRNA m6A modification by forming a stable m6A methyltransferase writer complex. Unlike DDX46, which inhibits innate immunity by entrapping m6A-demethylated antiviral transcripts in the nucleus (Zheng et al, 2017), DDX5 suppresses bacterial-induced inflammation by regulating the building of an mRNA m6A 'writer'. Indeed, stimulation with TLR2/4 agonists, which mimicked bacterial infection, increased the m6A level in DDX5-overexpressing cells but decreased it in DDX5 knockdown cells. An earlier study

reported that TLR2 and TLR4 signaling was regulated through promoting the transcription of the TLR2 and TLR4 genes by different histone modifications of epigenetic regulation (Shelke et al, 2022). In addition, METTL3-mediated m6A mRNA methylation was found to regulate neutrophil activation by targeting TLR4 signaling pathway (Luo et al, 2023). However, the precise mechanisms of m6A modification in bacterial infection was not investigated. In our study, TLR4 and TLR2 transcripts were themselves m6A-modified following LPS, FSL-1, or Pam3CSK4 treatment in a DDX5-dependent manner; this is the first study to

## Homeostasis

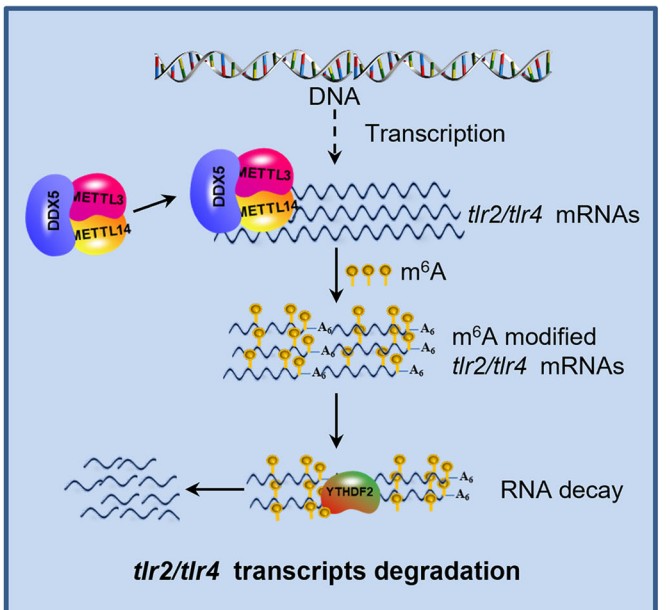

## Bacterial infection

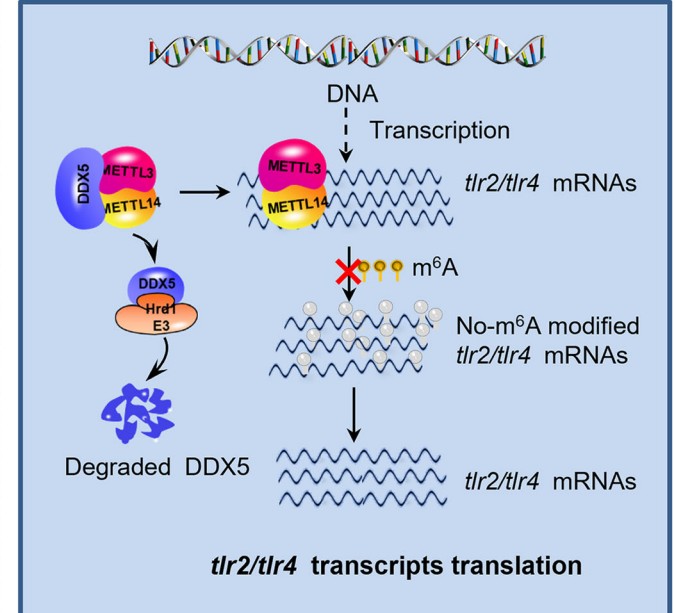

**Figure 8. Schematic diagram of the DDX5-mediated switch of the inflammatory response upon bacterial infection.**

In the absence of bacterial infection, DDX5/METTL3/METTL14 form a ternary complex. The complex promotes the m6A modification of TLR2/4 transcripts and affects the stability of transcripts, leading to their degradation via YTHDF2-dependent mRNAs decay, resulting in reduced expression of TLR2/4, which sequentially leads to the inactivation of the NF-κB-mediated inflammatory response and maintains homeostasis. Upon infection with pathogenic bacteria, DDX5 is degraded via the Hrd1-mediated ubiquitin-proteasome pathway, which prevents the formation of the mRNA methyltransferase ternary complex to specifically restrain m6A modification on TLR2/4 transcripts. TLR2/4 mRNAs are thus not degraded, but are excessively expressed, activating the NF-κB-mediated release of cytokines, such as IL-6, ultimately boosting the inflammatory response.

report that the TLR2/TLR4 transcripts could be regulated by post-transcriptional epigenetic modifications during bacterial infection. Previously, m6A modification of transcripts was associated with mRNA degradation (Zaccara and Jaffrey, 2020). Here, we found that DDX5 or METTL3 reduced the transcription and expression of TLR2/4 mRNAs. Consistent with the YTHDF2-dependent degradation of m6A-modified transcripts (Yue et al, 2015; Zaccara and Jaffrey, 2020), we found that DDX5 promoted the decay of TLR2/4 mRNA via YTHDF2, thus preventing the release of IL-6 or TNF-α. Our findings differ from those reporting that METTL3 initiates m6A modifications on TLR4 mRNA to promote its protein expression during neutrophil activation (Luo et al, 2023), suggesting that we discovered a novel function for the DDX5-METTL3-METTL14 complex via YTHDF2 to regulate the transcription of TLR2/4 mRNA during bacterial infection, which sheds light on mechanisms regulating the switch between inflammation and homeostasis.

Primary BMDM are naïve macrophages isolated from bone marrow, the birthplace of macrophages in blood and tissues of the whole body, which have been widely used in studies on innate immunity and TLR2/4-mediated NF-κB activation upon bacterial infection (Kondylis et al, 2017; Karmakar and Mandal, 2021). In the present study, we detected the expression of TLR2/4 in BMDM from DDX5$^{+/+}$ and DDX5$^{+/-}$ mice subjected to LPS, FSL-1, or Pam3CSK4 treatment. A clear increase in TLR2/4 and IL-6 or TNF-α production were observed in DDX5$^{+/-}$ mice compared with DDX5$^{+/+}$ mice. These results confirmed that DDX5 inhibited

bacterial inflammation. Inflammatory cytokines regulate both tissue-level and systemic adaptations to anticipated physical activity that exceeds homeostatic demands (Chovatiya and Medzhitov, 2014; Meizlish et al, 2021). Therefore, we verified the regulation of inflammatory signaling by DDX5 in vivo using knockout mice infected with pathogenic bacteria. Compared with wide type DDX5$^{+/+}$ mice, much higher expression of TLR2/4 was observed in the lungs of DDX5$^{+/-}$ mice, along with higher inflammatory cytokine levels, and more severe tissue inflammation. Besides macrophages, type 2 alveolar epithelial cells also participate in the inflammatory response by secreting IL-6 or TNF-α in the lungs (Meizlish et al, 2021). However, we have proven our findings in primary naïve BMDM, the primary cell type producing inflammatory cytokines during bacterial infection. Moreover, our findings ultimately verify that DDX5 blocks an inflammatory response after bacterial infection via METTL3-mediated m6A modification. However, several studies have reported that the m6A modification affected the inflammatory response in different diseases (Zheng et al, 2017; Jian et al, 2020; Mapperley et al, 2021; Wang et al, 2021); therefore, we further confirmed the specificity for DDX5-mediated m6A-modified regulation of inflammatory response in *Mettl3* cKO mice after infection with pathogenic bacteria. Our results showed that compared with wild type *Mettl3* WT mice, much higher expression of TLR2/4 and p-p65 was observed in primary BMDM or lungs of *Mettl3* cKO mice, along with higher inflammatory cytokine levels, more severe

tissue inflammation, and lower bacteria loading in the lungs of *Mettl3* cKO mice. All these results clearly indicate that DDX5 is a novel inflammation switch, which specifically modulates m6A levels of TLR2/4 transcripts during bacterial infection.

Although our previous study found that DDX5 is involved in m6A modification during antiviral innate immunity (Xu et al, 2021a; Xu et al, 2021b), its roles in bacterial infection remained unclear. Based on our results, we now propose a novel model for bacterial inflammation regulation by DDX5-mediated m6A modification that is presented in Fig. 8. In the absence of bacterial infection, DDX5 forms a complex with METTL3 and METTL14, which regulates target transcripts, including TLR2/4 mRNAs, through m6A modification. The m6A-modified TLR2/4 transcripts exported from the nucleus are recognized by YTHDF2, which facilitates their decay and prevents their translation and expression on the cell membrane. Ultimately, the absence of TLR2/4 receptors on the cell surface impairs TLR2/4-mediated inflammatory responses. In contrast, when cells encounter pathogenic bacteria, DDX5 is degraded through the Hrd1-mediated ubiquitin-proteasome pathway, preventing the formation of the complex and blocking the m6A modification of TLR2/4 transcripts. As a result, the transcription and expression of TLR2/4 mRNAs is possible, and translated TLR2/4 localizes to the cell membrane, where they recognize bacteria or PAMPs, thus activating the TLR2/4-mediated inflammatory response. This study not only demonstrates that DDX5 is a critical regulator of inflammation upon bacterial infection, but also highlights the molecular basis of the switch between inflammation and homeostasis. Our findings will enhance the overall understanding of inflammation during bacterial infection, thus facilitating the development of targeted anti-inflammatory therapies.

# Methods

## Reagents and tools

see Table 1.

**Table 1.  Reagents and tools.**

| Reagent/resource | Reference or source | Identifier or catalog number |
|---|---|---|
| Experimental models | | |
| DDX5+/- C57BL/6J (*Mus musculus*) | Cyagen Biosciences Inc. | N/A |
| METTL3$^{flox/+}$ C57BL/6J (*Mus musculus*) | Cyagen Biosciences Inc. | N/A |
| B6-CAG-CreERT2 (*Mus musculus*) | Cyagen Biosciences Inc. | N/A |
| Unmodified mouse embryonic fibroblasts | ATCC | NIH/3T3 |
| Mouse macrophages | ATCC | J774A.1 |
| *P. multocida* | CVCC | HuN001 |
| *S. aureus* | ATCC | 6538 |
| *M. pneumoniae* | ZhiXin Feng (Yangzhou University, China) | M129 |

**Table 1.**   (continued)

| Reagent/resource | Reference or source | Identifier or catalog number |
|---|---|---|
| Recombinant DNA | | |
| pEGF-N1 | TIANDZ | 60908-2680 |
| p3×FLAG-CMV-10 | zomanbio | ZK26130 |
| pCMV-Myc | TIANDZ | 60908-2050 |
| pCMV-HA | TIANDZ | 60908-2020 |
| IL-6 luciferase reporter | Jinhai Huang (Tianjin University, China) | N/A |
| pRL-TK Vector | Promega | E2231 |
| Antibodies | | |
| Mouse anti-METTL14 monoclonal antibody | Abcam | ab220030 |
| Rabbit anti-TLR2 polyclonal antibody | Abcam | a11225 |
| Mouse anti-TLR4 monoclonal antibody | Abcam | ab22048 |
| Rabbit anti-YTHDF2 monoclonal antibody | Abcam | ab220163 |
| Rabbit anti-DDX5 polyclonal antibody | Abcam | ab21696 |
| Rabbit anti-DDX5 polyclonal antibody | Abcam | ab126730 |
| Mouse anti-DDX5 monoclonal antibody | Abcam | ab53216 |
| Goat anti-DDX5 polyclonal antibody | Abcam | ab10261 |
| Rabbit anti-METTL3 monoclonal antibody | Abcam | ab195352 |
| Rabbit phospho-p65 (S276) monoclonal antibody | Abcam | ab183559 |
| Rabbit anti-m6A polyclonal antibody | Abcam | ab151230 |
| Anti-TICAM2 antibody | Abcam | ab17221 |
| Mouse anti-m6A monoclonal antibody | Abcam | ab208577 |
| Goat anti-rabbit IgG polyclonal antibody | Abcam | ab182016 |
| Donkey anti-mouse IgG H&L (Alexa Fluor® 488) | Abcam | ab150105 |
| Donkey anti-mouse IgG H&L (Alexa Fluor® 647) | Abcam | ab150107 |
| Donkey anti-rabbit IgG H&L (Alexa Fluor® 647) | Abcam | ab150075 |
| Donkey anti-rabbit IgG H&L (Alexa Fluor® 488 | Abcam | ab150073 |
| Donkey anti-goat IgG H&L (Alexa Fluor® 568) | Abcam | ab175474 |
| Mouse anti-METTL3 monoclonal antibody | Proteintech | 67733-1 |
| K48-linkage specific polyubiquitin rabbit polyclonal antibody | ABclonal Inc. | A18163 |
| Rabbit anti-HA-Tag polyclonal antibody | ABclonal Inc. | AE036 |
| Rabbit anti-Hrd1 polyclonal antibody | ABclonal Inc. | A2605 |
| Myc-Tag rabbit monoclonal antibody | ABclonal Inc. | AE070 |
| Mouse anti-HA-Tag monoclonal antibody | ABclonal Inc. | AE008 |
| Mouse anti-Myc-Tag monoclonal antibody | ABclonal Inc. | AE010 |
| Mouse anti-DDDDK-Tag monoclonal antibody | ABclonal Inc. | AE005 |
| Rabbit anti-METTL14 polyclonal antibody | ABclonal Inc. | A8530 |
| Mouse anti-β-actin monoclonal antibody | ABclonal Inc. | AC004 |
| Rabbit anti-TLR4 polyclonal antibody | ABclonal Inc. | A5258 |
| MyD88 Rabbit pAb | ABclonal Inc. | A0980 |

**Table 1.** (continued)

| Reagent/resource | Reference or source | Identifier or catalog number |
|---|---|---|
| Rabbit anti-METTL3 polyclonal antibody | ABclonal Inc. | A8370 |
| Horseradish peroxidase (HRP)-conjugated goat anti-mouse IgG (H+L) secondary antibody | Thermo Fisher Scientific | 31430 |
| HRP-conjugated goat anti-rabbit IgG (H+L) secondary antibody | Thermo Fisher Scientific | 31460 |
| Oligonucleotides and other sequence-based reagents | | |
| NC siRNA (Sense) 5'-UUCUCCGAACGUGUCACGUTT-3' | GenePharma | N/A |
| NC siRNA (Anti-sense) 5'-ACGUGACACGUUCGGAGAATT-3' | GenePharma | N/A |
| METTL3 siRNA (Sense) 5'-CCUCCAAGAUGAUGCACAUTT-3' | GenePharma | N/A |
| METTL3 siRNA (Anti-sense) 5'-AUGUGCAUCAUCUUGGAGGTT-3' | GenePharma | N/A |
| DDX5 siRNA (Sense) 5'-GCACAAUGGUAUGAACCAATT-3' | GenePharma | N/A |
| DDX5 siRNA (Anti-sense) 5'-UUGGUUCAUACCAUUGUGCTT-3' | GenePharma | N/A |
| METTL14 siRNA (Sense) 5'-GGGAGAGAUAGCACUAUCATT-3' | GenePharma | N/A |
| METTL14 siRNA (Anti-sense) 5'-UGAUAGUGCUAUCUCUCCCTT-3' | GenePharma | N/A |
| YTHDF2 siRNA (Sense) 5'-GGUAGCACAGAAGUUGCAATT-3' | GenePharma | N/A |
| YTHDF2 siRNA (Anti-sense) 5'-UUGCAACUUCUGUGCUACCTT-3' | GenePharma | N/A |
| Hrd1 siRNA (Sense) 5'-CUGUGACAGAUGCCAUCAUTT-3' | GenePharma | N/A |
| Hrd1 siRNA (Anti-sense) 5'-AUGAUGGCAUCUGUCACAGTT-3' | GenePharma | N/A |
| MyD88 siRNA (Sense) 5'-GAAGCGACUGAUUCCUAUUTT-3' | GenePharma | N/A |
| MyD88 siRNA (Anti-sense) 5'-AAUAGGAAUCAGUCGCUUCTT-3' | GenePharma | N/A |
| TRIF siRNA (Sense) 5'- GCAACAAAGUAUAUGGAAATT-3' | GenePharma | N/A |
| TRIF siRNA (Anti-sense) 5'- UUUCCAUAUACUUUGUUGCTT-3' | GenePharma | N/A |
| TLR2 forward primer 5'-GGAGTCAGACGTAGTGAGCG-3' | Sangon Biotech | N/A |
| TLR2 reverse primer 5'-AAATGCTGGGAGAACGAGCA-3' | Sangon Biotech | N/A |
| TLR4 forward primer 5'-GCAGGTGGAATTGTATCGCC-3' | Sangon Biotech | N/A |
| TLR4 reverse primer 5'-TTGCTCAGGATTCGAGGCTT-3' | Sangon Biotech | N/A |
| TAK1 forward primer 5'-ACAGGCACAAGCCAGATTGA-3' | Sangon Biotech | N/A |
| TAK1 reverse primer 5'- CTGGTAGGCGGACAAGAATCC-3' | Sangon Biotech | N/A |
| GAPDH forward primer 5'-AGGTCGGTGTGAACGGATTTG-3' | Sangon Biotech | N/A |
| GAPDH reverse primer 5'-TGTAGACCATGTAGTTGAGGTCA-3' | Sangon Biotech | N/A |
| Chemicals, enzymes, and other reagents | | |
| Lipofectamine 3000 DNA transfection reagents | Thermo Fisher Scientific | L3000015 |
| Lipofectamine RNAi MAX transfection reagent | Thermo Fisher Scientific | 13778150 |

**Table 1.** (continued)

| Reagent/resource | Reference or source | Identifier or catalog number |
|---|---|---|
| Pierce protein A/G magnetic beads | Thermo Fisher Scientific | 88803/26162 |
| LPS | InvivoGen | tlrl-pb5lps |
| FSL-1 | InvivoGen | tlrl-fsl |
| Pam3CSK4 | InvivoGen | tlrl-pms |
| Actinomycin D | Sigma-Aldrich | A1410 |
| iTaq Universal SYBR Green Supermix | Bio-Rad | 1725122 |
| Dynabeads™ M-280 streptavidin | Thermo Fisher Scientific, | 11205D |
| Anti-rabbit IgG-conjugated protein G dynabeads | Thermo Fisher Scientific | 10004D |
| Protease inhibitor cocktail | Roche | 04693116001 |
| MG132 | Selleck | S2619 |
| Tamoxifen | Sigma-Aldrich | 10540-29-1 |
| Software | | |
| GraphPad Prism 5.0 | https://www.graphpad.com/ | |
| Image J | https://imagej.net/ij/ij/index.html | |

## Ethics statement

All mice care procedures and experiments were approved by the Committee for the Ethics of Animal Experiments of the Lanzhou Veterinary Research Institute at the Chinese Academy of Agricultural Sciences (LVRIAEC-2022-003).

## Dual-luciferase reporter assay

MEFs were cultured then co-transfected with IL-6 luciferase reporter, pRL-TK plasmid and Myc-DDX5 plasmid or control vector with Lipofectamine 2000 transfection reagent. After 24 h, MEFs were treated with LPS, FSL-1, and Pam3CSK4 for 4 or 6 h and then cells were lysed for the detection of luciferase activity. The luciferase activity was measured using the DLR Assay System (Promega) with a Dual-Luciferase Reporter assay kit (Promega, Madison, WI, USA).

## RNA interference, plasmids, and transfection

MEFs were transfected with siRNA (10-20 nM) using Lipofectamine RNAi MAX transfection reagent and cultured for 36–48 h. Mouse macrophages were transfected with siRNA (40–60 nM) using Lipofectamine RNAi MAX transfection reagent and cultured for 48–72 h. In either case, the cells were lysed, and proteins were separated by SDS-PAGE and detected by Western blotting analysis. Sequences for siRNAs were showed in Reagents and Tools Table. The mouse DDX5 (ID: 13207), METTL3 (ID: 56335), METTL14 (ID: 210529), Hrd1 (ID: 74126), and Hrd1 (C329S) genes were sub-cloned into the pEGFP-N1, p3×FLAG-CMV-10, pCMV-Myc and pCMV-HA expression vectors. The plasmids were then transformed into Escherichia coli T1 cells and extracted using a plasmid kit according to the manufacturer's instructions. The plasmids were transfected into MEFs or 293T cells using Lipofectamine 3000 DNA transfection reagents for 24 h, after which the cells were lysed in

RIPA or NP40 buffer, and the lysates were used for Co-IP or western blotting.

## RNA decay assay

RNA decay assay was performed as described previously with modifications (Shen et al, 2016). Briefly, MEFs were transfected with plasmids encoding for Flag-DDX5 and control plasmids or siDDX5 and NC siRNA using Lipofectamine transfection reagents. Next, cells were treated with 5 μM actinomycin D for 0, 4, 6, and 8 h, followed by LPS (4 μg/mL), FSL-1 (50 ng/mL), and Pam3CSK4 (50 ng/mL) for another 4 h. Finally, the cells were collected to extract total RNA and measure TLR2/4 mRNA levels by qPCR.

## m6A sequencing and iCLIP-qPCR

Sequencing of m6A-modified transcripts was carried out as described previously (Kaczynski et al, 2019). Briefly, MEFs were transfected with siDDX5 or NC siRNA for 24–36 h and stimulated with LPS, FSL-1, or Pam3CSK4 for 6 h. Next, the mRNA extracted to capture m6A transcripts according to previous protocols (Kaczynski et al, 2019). Two independent biological replicates were employed for m6A sequencing. For iCLIP-qPCR, MEFs were transfected with Flag-DDX5/Flag-METTL3/control plasmids or siDDX5/siMETTL3/NC siRNA for 24–36 h, and the cells were stimulated with LPS, FSL-1, or Pam3CSK4 for 6 h. Then the cells were collected for the iCLIP-qPCR with anti-rabbit METTL3/YTHDF2 or anti-rabbit IgG-conjugated protein G dynabeads.

## Pull-down assay, co-immunoprecipitation (Co-IP), and mass spectrometry

For the Hrd1 pull-down assay, MEFs were seeded and cultured on 60-mm dishes for 24 h and lysed with NP40 lysis buffer containing protease inhibitor cocktail (Roche, Basel, Switzerland); the pull-down assay was performed according to a previous study (Xu et al, 2021b). The specific protein bands were separated from the pull-down assay and sequenced by Q Exactive plus combined with Ultimate 3000nano (Thermo Scientific, MA, USA). 293T cells, MEFs or mouse macrophages (Mø) were seeded on 60-mm dishes and co-transfected with plasmids, then the cells were lysed and precipitated with an antibody using protein G/A-magnetic beads. The beads were washed with cold phosphate-buffered saline and eluted with sodium dodecyl sulfate (SDS) loading buffer upon boiling for 10 min. Proteins isolated from the beads and cell lysates were separated by SDS-polyacrylamide gel electrophoresis (PAGE) and analyzed by western blotting using specific antibodies.

## In vivo ubiquitination assays

MEFs or Mø were treated with TLR2/4 agonists (LPS/FSL-1/Pam3CSK4), then the cells were lysed with NP40 lysis buffer containing protease inhibitor cocktail (Roche, Basel, Switzerland). The lysates underwent Co-IP based on the anti-DDX5 antibody, then the level of ubiquitination of DDX5 was detected immunoblot with K48-linkage specific polyubiquitin rabbit polyclonal antibody. The β-Actin served as a reference control. For verification of ubiquitination of DDX5 via Hrd1, MEFs were transiently transfected with FLAG-tag, FLAG-tagged Hrd1 or its E3 ligase-dead mutant C329S, and were lysed with NP40 lysis buffer containing protease inhibitor cocktail. The cell lysates underwent Co-IP with anti-DDX5 antibody-coated beads, followed by immunoblot analysis to detect K48 or K63-specific polyubiquitin of DDX5 and indicated proteins, respectively.

## Measurement of IL-6 and TNF-α release

MEFs and mouse macrophages were seeded on a 24-well plate and transfected with DDX5, METTL3, and YTHDF2 using Lipofectamine 3000 DNA transfection reagents for 24 h, or with siDDX5, siMETTL3, siYTHDF2, and NC siRNA using Lipofectamine RNAi MAX transfection reagent for 48 h. Next, the cells were infected with *P. multocida* (multiplicity of infection [MoI] = 20), *M. pneumoniae* (MoI = 1000), and *S. aureus* (MoI = 20) for 4 h or they were treated with LPS, FSL-1, and Pam3CSK4 for 4 h. The supernatant from each culture was collected to measure the production of IL-6 using a mouse IL-6 enzyme-linked immunosorbent assay (ELISA) kit (SEKM-0007; Solarbio, Beijing, China) according to the manufacturer's instructions. DDX5$^{+/+}$ mice, DDX5$^{+/-}$ mice, *Mettl3* cKO mice, and *Mettl3* WT mice were infected with *P. multocida* ($5 \times 10^3$ CFU/g) and *S. aureus* ($5 \times 10^5$ CFU/g) for 6, or with *M. pneumoniae* ($5 \times 10^5$ CFU/g) for 24 and 36 h. After that, mouse serum was collected to measure the production of IL-6 with the same ELISA kit as above. At the same time, mouse lungs were collected to measure the production of IL-6, as detailed above, and TNF-α using a mouse TNF-α ELISA kit (SEKM-0034; Solarbio, Beijing, China).

## Immunofluorescence analysis and confocal imaging

The cells were seeded on coverslips in 12-well plates and cultured for 18–24 h, followed by different treatments. Next, the cells were fixed and probed with mouse antibodies, followed by incubation with donkey anti-rabbit IgG H&L (Alexa Fluor® 647) and donkey anti-mouse IgG H&L (Alexa Fluor® 488). The nuclei were counterstained with 4′,6-diamidino-2-phenylindole (DAPI), and the cells were observed with a confocal laser scanning microscope (Nikon, Tokyo, Japan). The fluorescence intensity was quantified, and the co-localization was analyzed with Image J software. The data were analyzed by using GraphPad Prism 5.

## Detection of mRNA m6A levels in MEFs

MEFs were seeded on 60-mm dishes and transfected with pCMV-Flag-DDX5 or pCMV-Flag (control) using Lipofectamine 3000 DNA transfection reagents and cultured for 24 h, or they were transfected with siDDX5 or NC siRNA using Lipofectamine RNAi MAX transfection reagent and cultured for 48 h. Next, the cells were treated with LPS (4 μg/mL), FSL-1 (50 ng/mL), and Pam3CSK4 (50 ng/mL) for 4 h. The cells were collected to measure mRNA m6A levels by nucleotide array using either the colorimetric m6A RNA Methylation Assay Kit (ab185912; Abcam) or a dot-blot of purified mRNA as described previously (Roy et al, 2015). Methylene blue was used as RNA reference control.

## In vivo RNA pull down assay

MEFs were seeded on 60-mm dishes and transfected with Biotin labeled m6A peak of TLR2/4 transcripts, then the cells were

crosslinked with 0.4 J/cm$^2$ of 254 nm UV light in Crosslinkers. Then, cells were lysed by NP40 buffer containing a protease inhibitor cocktail and RNase inhibitor, the lysates were precipitated with an Dynabeads™ M-280 streptavidin. The beads were washed with cold by NP40 buffer and eluted with sodium dodecyl sulfate (SDS) loading buffer upon boiling for 10 min. Proteins isolated from the beads and cell lysates were separated by SDS-PAGE and analyzed by western blotting using METTL3 antibody and β-actin antibody.

## Immunoblot analysis

MEFs, mouse macrophages, 293T cells, and mouse lung tissue were lysed in RIPA or NP40 buffer containing a protease inhibitor cocktail (Roche, Basel, Switzerland), the lysates were centrifuged (12,000 rpm) for 10 min at 4 °C, and total proteins were quantified with a BCA Protein Assay Kit (Thermo Fisher Scientific). Samples were separated by 12% SDS-PAGE (Bio-Rad), blotted onto poly-vinylidene fluoride membranes, and immunoblotted with primary antibodies: anti-mouse DDX5 (1:2000), anti-rabbit DDX5 (1:10,000), anti-mouse METTL3 (1:1000), anti-rabbit METTL3 (1:1000), anti-METTL14 (1:1500), anti-β-actin (1:10,000), anti-p-P65 (1:2000), K48-linkage-specific polyubiquitin rabbit (1:1000), anti-mouse Myc (1:1000), anti-rabbit Myc (1:1000), anti-mouse Flag (1:1000), anti-mouse HA (1:2000), anti-rabbit HA (1:2500), anti-rabbit TLR2 (1:1000), anti-rabbit TLR4 (1:1000), anti-rabbit YTHDF2 (1:1000), MyD88 Rabbit pAb (1:1000), Anti-TICAM2 antibody (1:1000), and anti-rabbit Hrd1 (1:1000). After incubation with primary antibodies, the membrane was blotted with the following HRP-conjugated secondary antibodies: goat anti-mouse IgG (H + L; 1:10,000) and goat anti-rabbit IgG (H + L; 1:10,000). Reactive bands were detected using a Bio-Rad imaging system.

## Preparation of DDX5 knockout mice, METTL3 conditional knockout mice, and isolation of primary BMDMs

DDX5$^{+/-}$ mice were generated by Cyagen Biosciences Inc. (Shanghai, China) using CRISPR/Cas9 technology. The generation of knockout offspring was carried out according to our previous study (Xu et al, 2021b). Founder mice were hybridized with WT C57BL6/J mice to generate the mice identified and used for experiments. Mettl3 conditional knockout mice were generated on the C57BL6/J background by Cyagen Biosciences (Suzhou, Jiangsu, China) using CRISPR/Cas9-based targeting and homology-directed repair. The second and third exons of the METTL3 gene were flanked by two loxP sites to generate METTL3$^{flox/+}$ mice, Mettl3$^{flox/+}$ mice were crossbred with wide type mice to generate Mettl3$^{flox/+}$ mice, then Mettl3$^{flox/+}$ mice were inbred to generate Mettl3$^{flox/flox}$ mice, and Mettl3$^{flox/flox}$ mice were crossbred with B6-CAG-CreERT2 mice to generate Mettl3$^{flox/+, Cre}$ mice, finally, the Mettl3$^{flox/+,Cre}$ mice were backcrossed with Mettl3$^{flox/flox}$ mice to generate Mettl3 conditional knockout mice (Mettl3$^{flox/ flox, Cre(+/-)}$ mice, Mettl3 cKO mice). Moreover, Mettl3$^{flox/ flox, Cre (-/-)}$ mice (Mettl3 WT mice) were used as the controls. The mice were treated with Tamoxifen (20 µg/ g) daily per the recommended dosages for up to 7 consecutive days before experiments. Mouse BMDM were isolated from DDX5$^{+/-}$, DDX5$^{+/+}$, *Mettl3* cKO, and *Mettl3* WT mice, and cultured for 7–10 days. After that, the cells were counted with a cell counter and cultured in RPMI 1640 medium supplemented with 10% FBS.

## Histopathological and immunohistochemistry analysis

Wild type DDX5$^{+/+}$ mice, knockout DDX5$^{+/-}$ mice, *Mettl3* cKO, and *Mettl3* WT mice were infected with *P. multocida* ($5 \times 10^3$ CFU/g) by intraperitoneal injection for 12 h, with *S. aureus* ($5 \times 10^5$ CFU/g) by nasal delivery for 12 h, and with *M. pneumoniae* ($5 \times 10^5$ CFU/g) by nasal delivery for 36 h. Then, the mice were euthanized, and the lungs were collected for histopathological and immunohistochemistry analyses. The lungs were fixed in 4% paraformaldehyde overnight, sectioned, and stained with hematoxylin and eosin or anti-TLR2/TLR4 antibodies. Tissue sections were visualized under a light microscope.

## Statistical analysis

All tests were performed in accordance with biostatistical requirements. Statistical differences between treated and control groups were determined by analysis of variance using SPSS software, version 18.0 (SPSS, Chicago, IL, USA). Graphs were generated using GraphPad Prism 5.0 (GraphPad Software, San Diego, CA, USA), whereby "ns" indicates no significant difference and *, **, and *** indicate statistically significant differences at $p < 0.05$, $p < 0.01$, and $p < 0.001$, respectively.

# Data availability

No primary datasets have been generated and deposited.

# Peer review information

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

## Acknowledgements

This work was supported by the grants from the Youth Innovation Program of Chinese Academy of Agricultural Sciences (grant no. Y2022QC27), funding from National Natural Science Foundation of China (grant nos. 32373017, 32373019, and 32202806), funding from the Key Program of the Natural Science Foundation of Gansu Province (grant no. 23JRRA560), funding from the LVRI Young Talent Program (award no. 020493), and funding from the LVRI Yuan-Heng Talent Program (award no. NKLS2020-119).

## Author contributions

**Jian Xu**: Conceptualization; Resources; Data curation; Formal analysis; Funding acquisition; Validation; Investigation; Methodology; Writing—original draft; Writing—review and editing. **Li-Yuan Liu**: Conceptualization; Data curation; Software; Supervision; Validation; Investigation; Visualization; Methodology; Writing—original draft. **Fei-Jie Zhi**: Software; Formal analysis; Investigation; Visualization; Writing—original draft; Project administration. **Yin-Juan Song**: Resources; Formal analysis; Supervision; Funding acquisition; Investigation; Visualization; Methodology; Project administration. **Zi-Hui Zhang**: Software; Formal analysis; Investigation; Visualization; Methodology. **Bin Li**: Software; Formal analysis; Supervision; Validation; Methodology; Project administration. **Fu-Ying Zheng**: Resources; Software; Validation; Investigation; Methodology. **Peng-Cheng Gao**: Resources; Data curation; Investigation; Methodology. **Su-Zi Zhang**: Software; Formal analysis; Investigation; Visualization; Methodology. **Yu-Yu Zhang**: Software; Supervision; Investigation; Visualization; Methodology. **Ying Zhang**: Data curation; Supervision; Validation; Methodology; Writing—original draft; Project administration. **Ying Qiu**: Resources; Validation; Investigation; Methodology. **Bo Jiang**: Data curation; Supervision; Validation; Investigation; Methodology. **Yong-Qing Li**: Conceptualization; Resources; Formal analysis; Supervision; Writing—original draft. **Chen Peng**: Conceptualization; Resources; Supervision; Investigation; Methodology; Writing—original draft; Writing—review and editing. **Yue-Feng Chu**: Conceptualization; Supervision; Funding acquisition; Validation; Investigation; Visualization; Writing—original draft; Writing—review and editing.

## Disclosure and competing interests statement

The authors declare no competing interests.

# Expanded View Figures

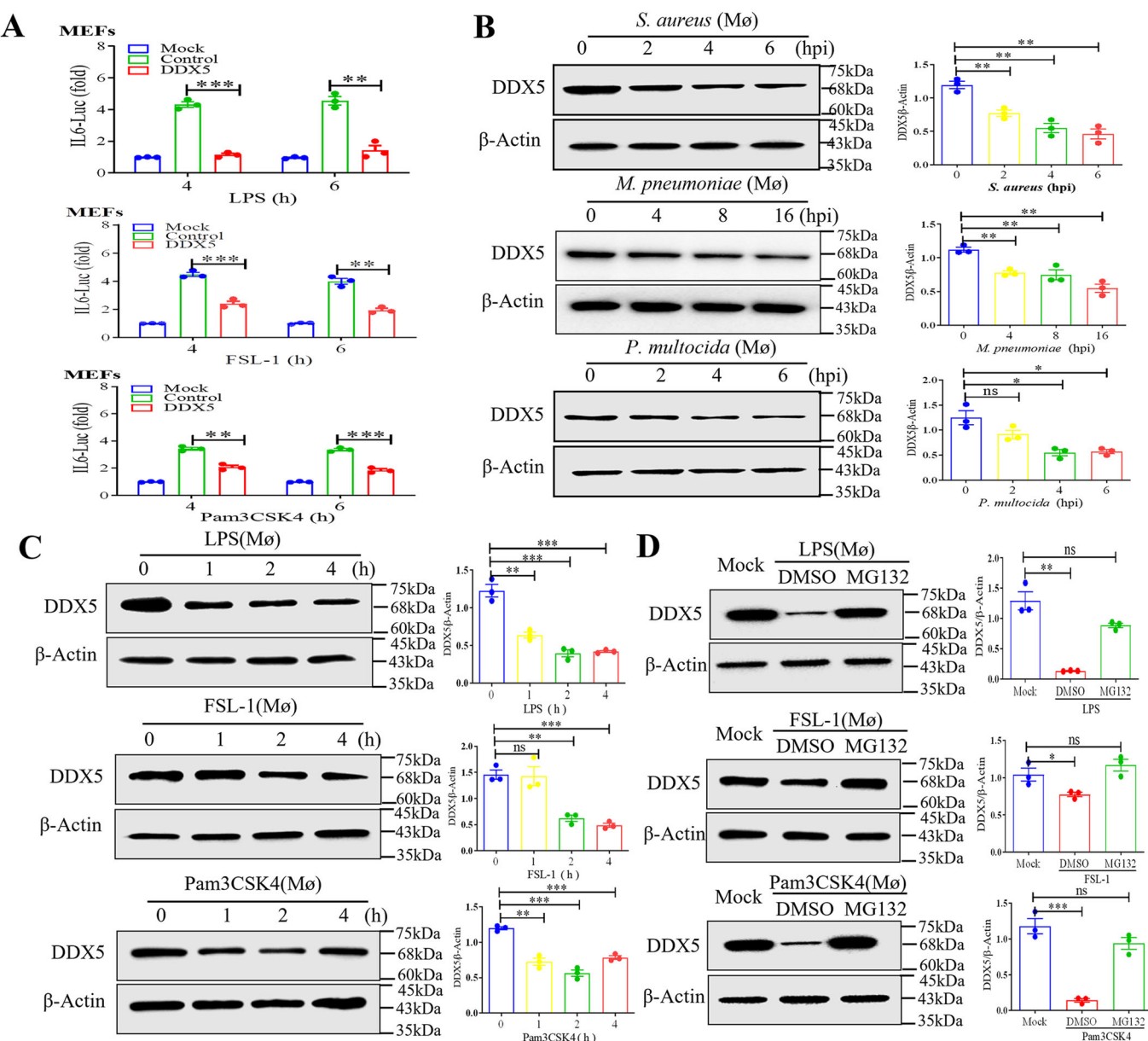

**Figure EV1. Luciferase activity of the IL-6 promoter in DDX5-overexpressing MEFs and the degradation of DDX5 via K48 linked ubiquitination in Mø treated with pathogenic bacteria or TLR2/4 agonists.**

(A) Luciferase activity of the IL-6 promoter in MEFs treated with LPS, FSL-1, and Pam3CSK4 for 4 and 6 h. (B) DDX5 expression in Mø infected with *S. aureus* for 0, 2, 4, and 6 h; *M. pneumoniae* for 0, 4, 8, and 16 h; *P. multocida* for 0, 2, 4, and 6 h. β-Actin as a reference control. (C) DDX5 expression in Mø treated with LPS, FSL-1, and Pam3CSK4 for 0, 1, 2, and 4 h. (D) DDX5 level in Mø treated with the MG132 proteasome inhibitor, as well as LPS, FSL-1, and Pam3CSK4 for 8 h; DMSO served as the control. The expression of DDX5 was quantified by the band intensity of DDX5/β-Actin in the western blot; the band intensity was measured by Image J software. Data information: In (A–D), all data are represented as the mean ± SEM of three biologically independent samples. "ns" indicates no significant difference, *$p < 0.05$, **$p < 0.01$, and ***$p < 0.001$ (Student's *t* test).

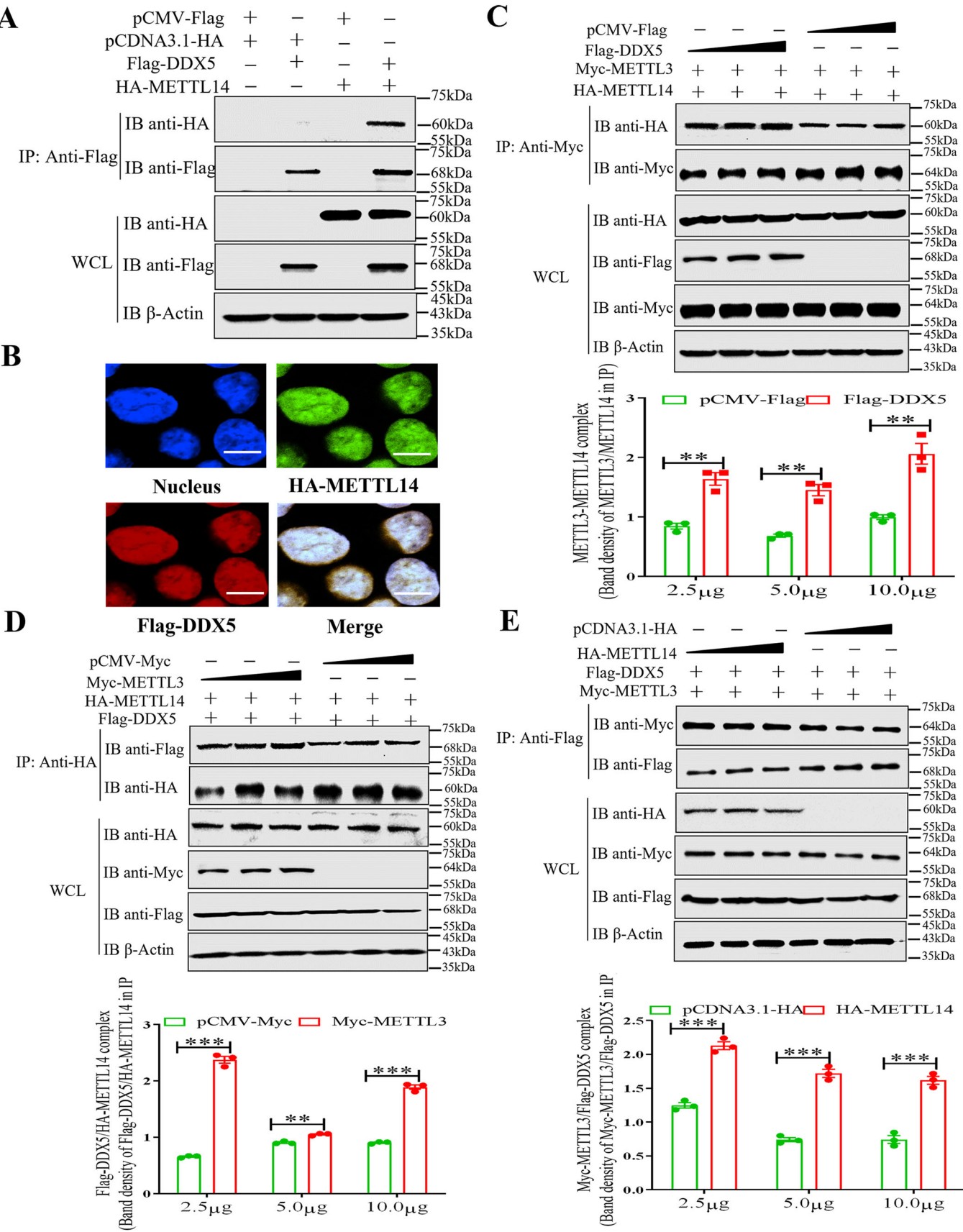

◀ **Figure EV2.  The interplay and stability of the DDX5-METTL3-METTL14 complex in 293T cells.**

(**A**) Interaction between Flag-DDX5 and HA-METTL14 in 293T cells detected by Co-IP. (**B**) Co-localization of HA-METTL14 and Flag-DDX5 in 293T cells. Scale bars: 10 μm. (**C–E**) The interplay of METTL3, METTL14, and DDX5 in 293T cells. Interaction between Myc-METTL3 and HA-METTL14 in 293T cells transfected with different doses of Flag-DDX5, Myc-METTL3, HA-METTL14 or control vector (2.5, 5.0, and 10.0 μg). The interaction of Myc-METTL3 and HA-METTL14 (**C**), HA-METTL14 and Flag-DDX5 (**D**), or Myc-METTL3 and Flag-DDX5 (**E**) was quantified by the band intensity of Myc-METTL3/HA-METTL14, Flag-DDX5/HA-METTL14, Myc-METTL3/ Flag-DDX5 in the IP system; the band intensity was measured by Image J software. Data information: In (**C–E**), all data are represented as the mean ± SEM of three biologically independent samples. $^{**}p < 0.01$ and $^{***}p < 0.001$ (Student's $t$ test).

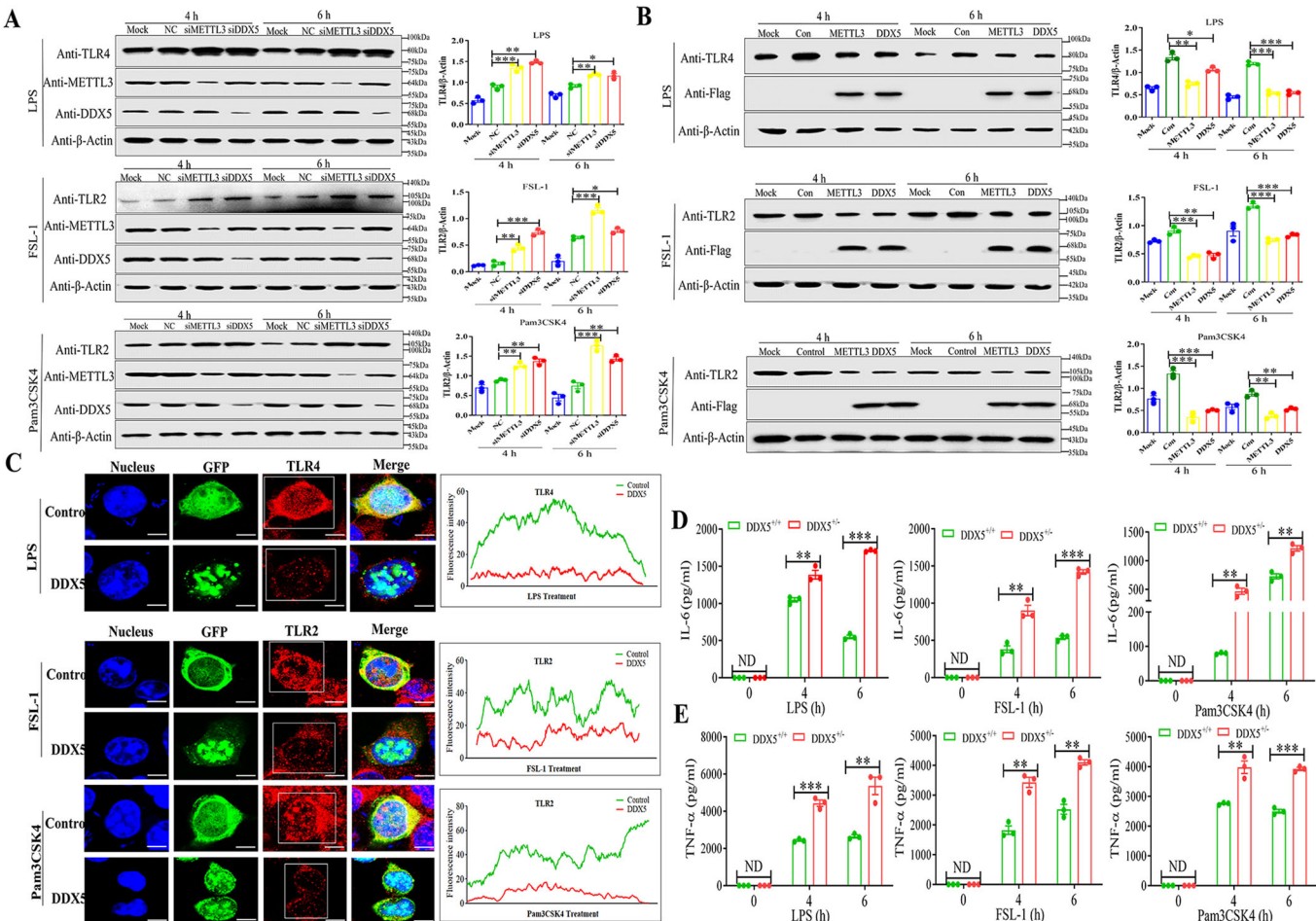

**Figure EV3.  mRNA m6A modification reduces the expression of TLR4/TLR2 in MEFs.**

(**A**, **B**) TLR4/TLR2 protein levels in NC, siDDX5/siMETTL3 MEFs (**A**), or in DDX5/METTL3- overexpressing MEFs (**B**) treated with LPS, FSL-1 and Pam3CSK4 for 4 and 6 h, respectively, and determined by western blot. The expression of TLR4 and TLR2 were quantified by the band intensity of DDX5/β-Actin in the western blot; the band intensity was measured by Image J software. (**C**) TLR2/4 localization and expression in GFP control or GFP-DDX5 expressing MEFs. MEFs were transfected with GFP control or GFP-DDX5 expressing plasmid for 24 h, then MEFs were treated with LPS, FSL-1, or Pam3CSK4, respectively; TLR4/TLR2 were detected in MEFs by IFA and CLSM. The green fluorescence indicated GFP (Control)/GFP-DDX5 (DDX5), and the red fluorescence indicated TLR4/TLR2. The right panels show the green line (Control) and red (DDX5) pixel intensity of the white markings in the pictures. Scale bars: 10 μm. (**D**, **E**) The production of IL-6 and TNF-α in primary BMDM following treatment with TLR2/4 agonists. IL-6 (I) or TNF-α (J) production in primary DDX5^+/+ BMDM or DDX5^+/- BMDM treated with LPS, FSL-1, or Pam3CSK4 for 4 and 6 h ($n = 3$). Data information: In (**A**, **B**, **D**, **E**), all data are represented as the mean ± SEM of three biologically independent samples. ND not detected. $*p < 0.05$, $**p < 0.01$, and $***p < 0.001$ (Student's $t$ test).

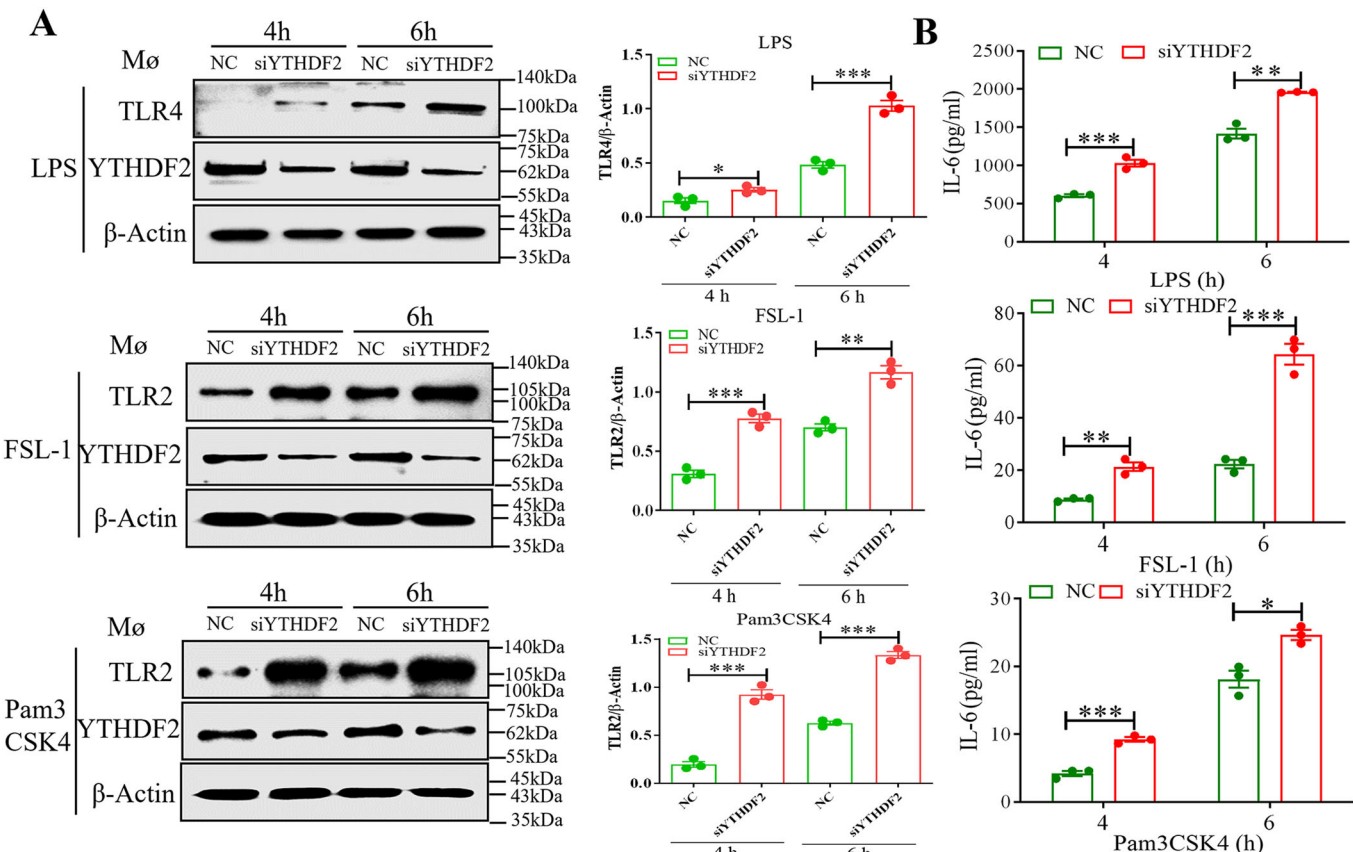

**Figure EV4.  Expression of TLR2/TLR4 and IL-6 production in YTHDF2 knockdown mouse macrophages (Mø) treated with LPS, FSL-1, and Pam3CSK4.**

(**A**) TLR4/TLR2 protein level in NC and siYTHDF2 mouse macrophages (Mø) treated with LPS, FSL-1, or Pam3CSK4 for 4 or 6 h. (**B**) IL-6 production in NC and siYTHDF2 mouse macrophages (Mø) treated with LPS, FSL-1, and Pam3CSK4. The expression of TLR4 and TLR2 were quantified by the band intensity of TLR4/β-Actin or TLR2/β-Actin in the western blot; the band intensity was measured by Image J software. Data information: In (**A**, **B**), all data are represented as the mean ± SEM of three biologically independent samples. *$p < 0.05$, **$p < 0.01$, and ***$p < 0.001$ (Student's $t$ test).

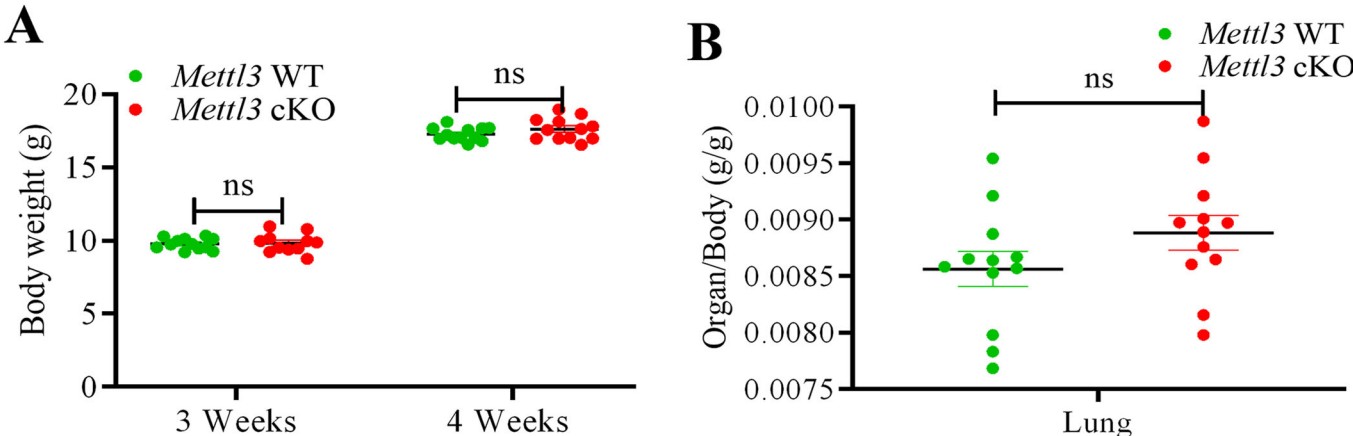

**Figure EV5. Quantification of the organ-body rate of *Mettl3* cKO and *Mettl3* WT mice.**

(**A**) Quantification of *Mettl3* WT or *Mettl3* cKO mice at postnatal 3 and 4 weeks ($n = 12$). (**B**) Organ-body rate was quantified between *Mettl3* WT and *Mettl3* cKO littermates ($n = 12$). Data information: In (**A**, **B**), all data are represented as the mean ± SEM of 12 biologically independent samples. "ns" indicates no significant difference (Student's *t* test).

