## [Peer Review File · EMBO Reports]

DDX5 inhibits inflammation by modulating m6A levels of TLR2/4 transcripts during bacterial infection

Jian Xu, Li-Yuan Liu, Fei-Jie Zhi, Yin-Juan Song, Zi-Hui Zhang, Bin Li, Fu-Ying Zheng, Peng-Cheng Gao, Su-Zi Zhang, Yu-Yu Zhang, Ying Zhang, Ying Qiu, Bo Jiang, Yong-Qing Li, Chen Peng, and Yue-Feng Chu

DOI: 10.15252/embr.202357416

Corresponding authors: Yue-Feng Chu (chuyuefeng@caas.cn) , Chen Peng (pengchenea@cau.edu.cn)

Review Timeline:

Submission Date:	29th Apr 23
Editorial Decision:	5th Jun 23
Authors' Correspondence:	11th Aug 23
Editor's Correspondence:	23 Aug 23
Authors' Correspondence:	6th Sep 23
Editor's Correspondence:	13th Sep 23
Appeal Received:	10th Oct 23
Editorial Decision:	20th Nov 23
Revision Received:	25th Nov 23
Editorial Decision:	12th Dec 23
Revision Received:	14th Dec 23
Accepted:	20th Dec 23

Editor: Achim Breiling

Transaction Report:

Dear Prof. Chu,

Thank you for the submission of your research manuscript to EMBO reports. I have now received the full set of referee reports that is copied below.

I am sorry to say that the decision on your manuscript is not a positive one. As you will see, referees #3 states that the conclusions are not fully supported by the data, in particular as important controls are missing, and also indicates novelty concerns. Referees #1 and #2 are more positive, but have also several concerns. Moreover, all referees mention technical shortcomings.

Given these comments, also indicating that the study does not provide the advance and broader impact we are looking for, and considering the amount of work required to address them, and finally the fact that EMBO reports can only invite revision of papers that receive strong support from the referees upon initial assessment, I cannot offer to publish your manuscript.

I am sorry to have to disappoint you this time. I nevertheless hope that the referee comments will be helpful in your continued work in this area, and I thank you once more for your interest in our journal.

Yours sincerely

Referee #1:

In this study, Xu et al investigated how DDX5 negatively regulated TLR2/4 mRNA at steady state via m6A of TLR2/4 transcripts (via YTHDF2-modulated RNA degradation), which was achieved by DDX5's interaction METTL3 and METTL14 forming an m6A writing complex. This process was reverted by bacterial infection and TLR stimulation that induced DDX5's recruitment to an ER-localized ubiquitinate Hrd1 followed by DDX5's degradation, resulting in disruption of the DDX5 m6A writing complex and the subsequent degradation of TLR2/4 mRNAs. Overall, they provided extensive and consistent data throughout the study. The experimental design was appropriate and well executed.

Major comments:

1. While they have proposed an interesting biochemical model for DDX5's role in TLR2/4 m6A modification and consequent degradation, it remains unclear how TLR stimulation led to the recruitment of DDX5 to Hrd1 for destruction of the DDX5 complex. Is it through the MyD88-dependent or independent pathway? Or this is through a feedback loop of downstream effector molecule. The authors may provide some insight about the receptor-proximal signaling in this downstream DDX5 effect.
2. While the authors made a good effort to examine the in vivo importance of this pathway, it is very hard to draw equation between DDX5 KO or METTL3 KO to the loss of m6A on TLR2/4 transcripts. I think that the strong biochemical and molecular data are sufficient as a complete story. If the authors really want an in vivo outcome, they would need to mutate the m6A site on TLR2/4, which might be challenging to do.

Minor comments:

1. The DDX5 dynamics seems to be different in response to stimulations of various TLR ligands. Specifically, DDX5 bounced back rather quickly in response to TLR2 ligand (e.g. Fig. EV3.)
2. It might be a good idea to quantify DDX5 protein levels or percentage of degradation so that one can be more objective in concluding the DDX5 degradation dynamics.
3. Is DDX5 also ubiquitinated by K63?
4. The authors had this subtitle in the paper "DDX5 halts the translation of TLR2/4 transcripts by promoting mRNA m6A modification". In this section, they stated "As shown in Fig 4G, the mRNA level of TLR2 was significantly higher in DDX5 or METTL3 knockdown MEFs treated with LPS (Fig 4G) but significantly lower in MEFs overexpressing either DDX5 or METTL3 upon LPS treatment (Fig 4H). Since the regulation was so dramatic at the RNA levels, it is not accurate for them to claim "halts the translation...".

Referee #2:

The authors have found that upon bacteria infection , DDX5 was recruited to ER to bind to HRD1 for degradation via the

ubiquitin-proteasome pathway. Using DDX5 het mice, siRNA knock down or over-expression strategies, the authors have proved that DDX5 can interact with METTL3/14 to form a m6A writing complex. This targets TLR2/4 mRNA for degradation. The authors have provided a substantial amount of data to demonstrate the molecule mechanism. The in vivo function showing how DDX5 het mice and Mettl3 KO mice affected inflammation and the outcome of bacteria infection is critical. The concept is interesting and the data are convincing which contain details of both molecular mechanism and the cellular or the in vivo function.

Comments:

1. Supplementary figure 1DEF and Supplementary figure 2A are important data showing the effects of different bacteria infection, which should be moved into the main figure. By contrast, some over-expression data using THP1 cell line could be moved into supplementary figure.
2. "significantly" was used many times in the manuscript. Overstatement should be avoided.
3. Language editing is needed throughout the manuscript.
4. RNA-seq analysis is suggested be performed using macrophages from DDX5 het mice and Mettl3 KO mice with or without bacterial infection. This might provide more information about whether other cytokines were also affected or how inflammation was regulated.

Some minor suggestions are listed below.

- 1, In the text and Figure 5, p in p65 should be written as a small letter. Some P65 were written as a big letter.
- 2, In figure 3 K and L: Two panels in K should be reversed, i.e. DDX5 over-expression should be in the left, DDX5 siRNA in the right. This can fit with Fig. 3K.
3. In the Model (right panel): mRNAs should be written as mRNA.

Referee #3:

The work by Xu, Liu and colleagues provides evidence for a role for the previously described DDX5-METTL3-METTL14 on the TLR2 and TLR4-mediated pro-inflammatory response (mostly IL-6 and TNF alpha levels) against bacterial components. Authors use bacterial infections as well as TLR2 and TLR4 agonists in two cellular models (MEFs and murine macrophages) as well as in in vivo models.

Previous studies (doi: 10.1371/journal.ppat.1009530 and doi: <https://doi.org/10.1128/spectrum.01098-22>) already reported the relevance of the DDX5-METTL3-METTL14 complex on the type-I IFN and pro-inflammatory responses during VSV or IAV replication in MEFs and A549 cells, respectively. This, through the m6A-mediated regulation of DHX58, p65, IKK gamma, IFN beta and/or IL-6 mRNAs. In this sense, the m6A-mediated regulation of TLR2 and TLR4 mRNAs by the DDX5-METTL3-METTL14 complex during bacterial infection and their impact on IL-6 levels reported in this study is not surprising. In addition, most figures lack a control condition, which impedes to obtain compelling conclusions.

Specific comments:

- Since data on Fig. 1 are mostly based on the analysis of IL-6 (secreted levels and promoter induction) in cell culture, it would be important to determine whether these observations are exclusive for this cytokine (or its mRNA) or whether also applies to other pro-inflammatory cytokine markers. This to decipher whether the effect of DDX5-METTL3-METTL14 complex is exerted on the IL-6 mRNA via m6A-YTHDF2 (as previously reported; doi: <https://doi.org/10.1128/spectrum.01098-22>) or on other players in the NF-kB pathway.
- From Fig. 2, it is unclear how TLR2 and TLR4 activation (by agonists or infection) results in reduced levels of DDX5 and how this relates to Hrd1-mediated ubiquitination and degradation of DDX5. ¿Is DDX5 relocated to the ER or is Hrd1 relocated to the nucleus during bacterial infection and treatment with TLR2 and TLR4 agonists?
- In my opinion, the most critical weakness of this work is the lack of a "Mock condition" (vehicle for the drugs) in all the experiments in which TLR2 and TLR4 agonists were used. Indeed, authors included this critical control only in some of them. This is critical because authors show that LPS, FSL-1 and Pam3CSK4 induce an important reduction of DDX5 and thus, all the observations and conclusions are performed under reduced levels of DDX5 that in some cases are further reduced by siRNA. See Figs. 3K, 3L, 4A-4K, 6A-6Q as examples.
- From Fig. 4, if authors propose that DDX5 is required for the stability of the METTL3/METTL14 complex, ¿why DDX5 knock down only affects the levels of m6A on the TLR2 and TLR4 mRNAs? Once again, in the absence of the vehicle control, all these conclusions arise from data obtained under reduced levels of DDX5 because the treatment with LPS, FSL-1 and Pam3CSK4 (used as a control) that were further decreased by siRNA.
- What was the m6A pattern of TLR4 mRNA under LPS treatment and in the TLR2 mRNA under FSL-1 and Pam3CSK4 treatments?

- Fig. 4A: why the m6A site in the LPS-NC condition has A/C and not only A as the others?

- The sub-title "DDX5 halts translation of TLR2/4 transcripts..." should be corrected as authors do not evaluate translation in their experiments.

** As a service to authors, EMBO Press provides authors with the ability to transfer a manuscript that one journal cannot offer to publish to another journal, without the author having to upload the manuscript data again. To transfer your manuscript to another EMBO Press journal using this service, please click on Link Not Available

August 11, 2023

Dear Editor,

Thank you and the reviewers' comments about our manuscript EMBOR-2023-57416V1.

We have carefully considered the reviewers' suggestion. Referees #1 and #2 had positive comments, but Referees #3 raised some concerns. We therefore performed more experiments and revised the manuscript to answer all of the reviewers' questions.

The major details were revised as described below:

- (1) The redistribution of DDX5 after treatment of TLR2/TLR4 agonists.
- (2) **The co-localization of DDX5 and Hrd1 in mouse macrophages (M ϕ) after treatment of TLR2/TLR4 agonists.**
- (3) **Degradation of DDX5 was not through K63 linked ubiquitin-proteasome pathway.**
- (4) **TNF- α production was detected *in vitro* and *in vivo* infected with pathogenic bacteria.**
- (5) **For illustrating the novelty of our study, we have stated that no studies so far have examined if RNA helicases contribute to bacterial inflammation or the molecular basis of inflammation quiescence during homeostasis in the introduction, results and discussion of revised version.**
- (6) **We have replenished all the lacking control or Mock conditions in all figures.**

We appreciate it if you could give us a chance to submit the revised manuscript.

Sincerely,

Yuefeng Chu, Ph.D.

State Key Laboratory for Animal Disease Control and Prevention,

College of Veterinary Medicine, Lanzhou University, Lanzhou Veterinary Research Institute, Chinese Academy of Agricultural Sciences, Lanzhou 730000, People's Republic of China.

Email: chuyuefeng@caas.cn (YC)

Phone: 86+0931-8342706

Dear Dr. Chu,

In order to evaluate this, could you please send me a complete and thorough point-by-point response to ALL the referee concerns with explanations how the individual points are or will be addressed in the revised manuscript?

Thanks,

Achim Breiling
Senior editor
EMBO reports

Authors' Correspondence

6th Sep 2023

Dear Dr. Achim Breiling,

Thanks a lot for evaluation of our manuscript. We have carefully prepared the point-by-point response to ALL the referee concerns with explanations, please find the file attached.

Looking forward to your positive response!

Best regards,

Yuefeng

Dr. Yuefeng Chu

State Key Laboratory of Veterinary Etiological Biology, College of Veterinary Medicine, Lanzhou University, Lanzhou Veterinary Research Institute, Chinese Academy of Agricultural Sciences, Lanzhou 730000 (P.R. China).

1 Xujiaping, Yanchangbu

Chengguan District

Lanzhou, Gansu 730046, China

Cell 86-13619391184

Dear Yuefeng,

Thank you for your letter and the preliminary p-b-p-response. If you feel that you can address the referee concerns and have data that would considerably strengthen the study (as outlined in their reports), please submit your fully revised paper as new submission including your final detailed point-by-point response and indicate during submission (and in your cover letter) that this is a re-submission, also mentioning the previous manuscript number and this conversation. Our assistant will then link this to the previous version.

Please note that all referee concerns must be addressed in the revised manuscript and in a detailed point-by-point response. I went through your preliminary p-b-p-response, and it seems the referee concerns will be adequately addressed. Nevertheless, Acceptance of your manuscript will depend on a positive outcome of a second round of review. It is EMBO reports policy to allow a single round of revision only and acceptance of the manuscript will therefore depend on the completeness of your responses included in the next, final version of the manuscript.

When submitting your revised manuscript, please also review the instructions that follow below to speed up the process.

1)) a .docx formatted version of the final manuscript text (including legends for main figures, EV figures and tables), but without the figures included. Figure legends should be compiled at the end of the manuscript text.

2) individual production quality figure files as .eps, .tif, .jpg (one file per figure), of main figures (up to 8) and EV figures (up to 5). Please upload these as separate, individual files upon re-submission.

The Expanded View format, which will be displayed in the main HTML of the paper in a collapsible format, has replaced the Supplementary information. You can submit up to 5 images as Expanded View. Please follow the nomenclature Figure EV1, Figure EV2 etc. The figure legend for these should be included in the main manuscript document file in a section called Expanded View Figure Legends after the main Figure Legends section. Additional Supplementary material should be supplied as a single pdf file labeled Appendix. The Appendix should have page numbers and needs to include a table of content on the first page (with page numbers) and legends for all content. Please follow the nomenclature

Appendix Figure Sx, Appendix Table Sx etc. throughout the text, and also label the figures and tables according to this nomenclature.

http://wol-prod-cdn.literatumonline.com/pb-assets/embosite/EMBOPress_Figure_Guidelines_061115-1561436025777.pdf

See also the guidelines for figure legend

preparation: <https://www.embopress.org/page/journal/14693178/authorguide#figureformat>

4) a complete author checklist, which you can download from our author guidelines (<https://www.embopress.org/page/journal/14693178/authorguide>). Please insert page numbers in the checklist to indicate where the requested information can be found in the manuscript. The completed author checklist will also be part of the RPF.

Please also follow our guidelines for the use of living organisms, and the respective reporting

guidelines: <http://www.embopress.org/page/journal/14693178/authorguide#livingorganisms>

5) that primary datasets produced in this study (e.g. RNA-seq, ChIP-seq, structural and array data) are deposited in an appropriate public database. If no primary datasets have been deposited, please also state this in a dedicated section (e.g. 'No primary datasets have been generated and deposited'), see below.

The accession numbers and database should be listed in a formal "Data Availability" section (placed after Materials & Methods) that follows the model below. This is now mandatory (like the COI statement). Please note that the Data Availability Section is restricted to new primary data that are part of this study. This section is mandatory. As indicated above, if no primary datasets have been deposited, please state this in this section

Data availability

Moreover, please note these editorial requests:

6) Our journal encourages inclusion of *data citations in the reference list* to directly cite datasets that were re-used and obtained from public databases. Data citations in the article text are distinct from normal bibliographical citations and should directly link to the database records from which the data can be accessed. In the main text, data citations are formatted as follows: "Data ref: Smith et al, 2001" or "Data ref: NCBI Sequence Read Archive PRJNA342805, 2017". In the Reference list, data citations must be labeled with "[DATASET]". A data reference must provide the database name, accession number/identifiers and a resolvable link to the landing page from which the data can be accessed at the end of the reference. Further instructions are available at: <http://www.embopress.org/page/journal/14693178/authorguide#referencesformat>

7) Regarding data quantification and statistics, please make sure that the number "n" for how many independent experiments were performed, their nature (biological versus technical replicates), the bars and error bars (e.g. SEM, SD) and the test used to calculate p-values is indicated in the respective figure legends (also for potential EV figures and all those in the final Appendix). Please also check that all the p-values are explained in the legend, and that these fit to those shown in the figure. Please provide statistical testing where applicable. Please avoid the phrase 'independent experiment', but clearly state if these were biological or technical replicates. Please also indicate (e.g. with n.s.) if testing was performed, but the differences are not significant. In case n=2, please show the data as separate datapoints without error bars and statistics. See also: <http://www.embopress.org/page/journal/14693178/authorguide#statisticalanalysis>

8) Please add scale bars of similar style and thickness to microscopic images, using clearly visible black or white bars (depending on the background). Please place these in the lower right corner of the images themselves. Please do not write on or near the bars in the image but define the size in the respective figure legend.

9) Please also note our reference format:

10) We updated our journal's competing interests policy in January 2022 and request authors to consider both actual and perceived competing interests. Please review the policy <https://www.embopress.org/competing-interests> and update your competing

interests if necessary. Please name this section 'Disclosure and Competing Interests Statement' and put it after the Acknowledgements section.

11) We now use CRediT to specify the contributions of each author in the journal submission system. CRediT replaces the author contribution section. Please use the free text box to provide more detailed descriptions and do not provide your final manuscript text file with an author contributions section. See also our guide to authors:

<https://www.embopress.org/page/journal/14693178/authorguide#authorshipguidelines>

12) We would encourage you to use 'Structured Methods', our new Materials and Methods format. According to this format, the Materials and Methods section should include a Reagents and Tools Table (listing key reagents, experimental models, software and relevant equipment and including their sources and relevant identifiers) followed by a Methods and Protocols section in which we encourage the authors to describe their methods using a step-by-step protocol format with bullet points, to facilitate the adoption of the methodologies across labs. More information on how to adhere to this format as well as downloadable templates (.doc or .xls) for the Reagents and Tools Table can be found in our author guidelines (section 'Structured Methods'):

Please make sure the manuscript sections are ordered like this, using these names:
Title page - Abstract – Keywords - Introduction - Results - Discussion - Materials and Methods - Data availability section - Acknowledgements - Disclosure and Competing Interests Statement - References - Figure legends - Expanded View Figure legends

Finally, please note that all corresponding authors are required to supply an ORCID ID for their name upon submission of a revised manuscript. Please find instructions on how to link the ORCID ID to the account in our manuscript tracking system in our Author guidelines: <http://www.embopress.org/page/journal/14693178/authorguide#authorshipguidelines>

Please let me know if you have further questions.

Best regards,

Achim

Point-by-point response to the referee concerns

We are highly grateful for the valuable and inspirational comments and suggestions from the reviewers, all of which have been fully addressed by adding additional experiments, explanations and discussion as well as figure re-arrangement. Point-by-point responses are below.

Referee #1:

In this study, Xu et al investigated how DDX5 negatively regulated TLR2/4 mRNA at steady state via m6A of TLR2/4 transcripts (via YTHDF2-modulated RNA degradation), which was achieved by DDX5's interaction METTL3 and METTL14 forming an m6A writing complex. This process was reverted by bacterial infection and TLR stimulation that induced DDX5's recruitment to an ER-localized ubiquitinate Hrd1 followed by DDX5's degradation, resulting in disruption of the DDX5 m6A writing complex and the subsequent degradation of TLR2/4 mRNAs. Overall, they provided extensive and consistent data throughout the study. The experimental design was appropriate and well executed.

Major comments:

1. While they have proposed an interesting biochemical model for DDX5's role in TLR2/4 m6A modification and consequent degradation, it remains unclear how TLR stimulation led to the recruitment of DDX5 to Hrd1 for destruction of the DDX5 complex. Is it through the MyD88-dependent or independent pathway? Or this is through a feedback loop of downstream effector molecule. The authors may provide some insight about the receptor-proximal signaling in this downstream DDX5 effect.

Answer: In order to explore whether the recruitment of DDX5 to Hrd1 upon TLR stimulation was MyD88-dependent or independent pathway, new experiments were performed. The synthesis of MyD88 and TRIF was depressed by siRNA in MEFs or M ϕ followed by TLR stimulation, and the interaction of DDX5 and Hrd1 was examined by Co-IP analysis. The results have been added in Fig EV2M and EV2N in the revised manuscript.

2. While the authors made a good effort to examine the in vivo importance of this pathway, it is very hard to draw equation between DDX5 KO or METTL3 KO to the loss of m6A on TLR2/4 transcripts. I think that the strong biochemical and molecular data are sufficient as a complete story. If the authors really want an in vivo outcome, they would need to mutate the m6A site on TLR2/4, which might be challenging to do.

Answer: We appreciate the comments and acknowledge that mutation of m6A site on TLR2/4 would provide direct evidence for the importance of DDX5 in inflammation regulation. As a result, we performed the following new experiments: Biotin-labeled TLR2/4 transcripts with or without mutations in m6A modification sites were synthesized and the interaction of TLR2/4 transcripts and METTL3 were examined by pull down assay. We observed that TLR2/4 with mutated m6A modification sites failed to interact with METTL3 upon stimulation, further confirming the critical role of METTL3 in the regulation of TLR2/4 transcripts. The results have been added in Fig EV3D and E in the revised version.

Fig EV3D

m6A Peak of TLR2/4 transcripts	Sequence
m6A tlr4	5'-CUACCAUUUUGACAUCAGUGGACUUUUUUCAUUCUCUGUCUUUCCAUAUU-3'
m6A tlr4 -M	5'-CUACCAUUUUGACAUCAGUGGBCUUUUUUCAUUCUCUGUCUUUCCAUAUU-3'
m6A tlr2-1	5'-AUGACUAAGAUGCAUCUGAUCAGGACAGGGGCAUGUUUAUAUGCAUGUC-3'
m6A tlr2-1 -M	5'-AUGACUAAGAUGCAUCUGAUCAGGBCAGGGGCAUGUUUAUAUGCAUGUC-3'
m6A tlr2-2	5'-UGGCACUGUGAGGACAGGGCCAGUAAUUUAGGGUA-3'
m6A tlr2-2 -M	5'-UGGCACUGUGAGGBCAGGGCCAGUAAUUUAGGGUA-3'

Fig EV3E

Minor comments:

1. The DDX5 dynamics seems to be different in response to stimulations of various TLR ligands. Specifically, DDX5 bounced back rather quickly in response to TLR2 ligand (e.g. Fig. EV3.)

Answer: We appreciate the comments and analyzed the band intensity of DDX5/ β -Actin by Image J software (Fig EV3), the expression of DDX5 was significantly decreased in all samples after Pam3CSK4 treatment. The results have been added to Fig. EV1C

2. It might be a good idea to quantify DDX5 protein levels or percentage of degradation so that one can be more objective in concluding the DDX5 degradation dynamics.

Answers: We appreciate the suggestion and quantified DDX5 protein levels or the percentages of DDX5 degradation in the revised version, the results were added to Fig. 2A-2C and EV1B-1D.

3. Is DDX5 also ubiquitinated by K63?

Answers: To determine if DDX5 is K63 ubiquitinated, we performed a new experiment and added the new data to Fig 2E. Our results showed that the K63 linked ubiquitination of DDX5 was not changed in LPS-stimulated macrophages.

4. The authors had this subtitle in the paper "DDX5 halts the translation of TLR2/4 transcripts by promoting mRNA m6A modification". In this section, they stated "As shown in Fig 4G, the mRNA level of TLR2 was significantly higher in DDX5 or METTL3 knockdown MEFs treated with LPS (Fig 4G) but significantly lower in MEFs overexpressing either DDX5 or METTL3 upon LPS treatment (Fig 4H). Since the regulation was so dramatic at the RNA levels, it is not accurate for them to claim "halts the translation...".

Answers: The sentence has been changed to " DDX5 restrained inflammatory responses via regulating m6A modification of TLR2/4 transcripts during bacterial infection. " in the revised version.

Referee #2:

The authors have found that upon bacterial infection, DDX5 was recruited to ER to bind to HRD1 for degradation via the ubiquitin-proteasome pathway. Using DDX5 het mice, siRNA knock down or over-expression strategies, the authors have proved that DDX5 can interact with METTL3/14 to form a m6A writing complex. This targets TLR2/4 mRNA for degradation. The authors have provided a substantial amount of data to demonstrate the molecule mechanism. The in vivo function showing how DDX5 het mice and Mettl3 KO mice affected inflammation and the outcome of bacteria infection is critical.

The concept is interesting and the data are convincing which contain details of both molecular mechanism and the cellular or the in vivo function.

Comments:

1. Supplementary figure 1DEF and Supplementary figure 2A are important data showing the effects of different bacteria infection, which should be moved into the main figure. By contrast, some over-expression data using THP1 cell line could be moved into supplementary figure.

Answers: We moved the Supplementary figure 1DEF and Supplementary figure 2A into figure 1, and some over-expression data were moved into supplementary figure in the revised version.

2. "significantly" was used many times in the manuscript. Overstatement should be avoided.

Answers: We have checked the whole text reduced the use of "significantly" in the revised version.

3. Language editing is needed throughout the manuscript.

Answers: We appreciate the suggestion and have performed language editing in the revised manuscript.

4. RNA-seq analysis is suggested be performed using macrophages from DDX5 het mice and Mettl3 KO mice with or without bacterial infection. This might provide more information about whether other cytokines were also affected or how inflammation was regulated.

Answers: We agree with the reviewer's suggestion. We have performed new experiments to detect TNF- α production in DDX5^{+/-} BMDM or DDX5^{+/-} BMDM, MEFs, macrophages (M ϕ) after LPS, FSL-1, or Pam3CSK4 treatment, as well as in mice after bacterial infection. We have added the results in Fig 1B, 1D and 1D, Fig EV3J and Fig 7 E, 7H and 7J in the revised manuscript.

Some minor suggestions are listed below.

1. In the text and Figure 5, p in p65 should be written as a small letter. Some P65 were written as a big letter.

Answers: changed as suggested.

2. In figure 3 K and L: Two panels in K should be reversed, i.e. DDX5 over-expression should be in the left, DDX5 siRNA in the right. This can fit with Fig. 3K.

Answers: revised as suggested.

3. In the Model (right panel): mRNAs should be written as mRNA.

Answers: changed as suggested.

Referee #3:

The work by Xu, Liu and colleagues provides evidence for a role for the previously described DDX5-METTL3-METTL14 on the TLR2 and TLR4-mediated pro-inflammatory response (mostly IL-6 and TNF alpha levels) against bacterial components. Authors use bacterial infections as well as TLR2 and TLR4 agonists in two cellular models (MEFs and murine macrophages) as well as in in vivo models.

Previous studies (doi: 10.1371/journal.ppat.1009530 and doi: <https://doi.org/10.1128/spectrum.01098-22>) already reported the relevance of the DDX5-METTL3-METTL14 complex on the type-I IFN and pro-inflammatory responses during VSV or IAV replication in MEFs and A549 cells, respectively. This, through the m6A-mediated regulation of DHX58, p65, IKK gamma, IFN beta and/or IL-6 mRNAs. In this sense, the m6A-mediated regulation of TLR2 and TLR4 mRNAs by the DDX5-METTL3-METTL14 complex during bacterial infection and their impact on IL-6 levels reported in this study is not surprising. In addition, most figures lack a control condition, which impedes to obtain compelling conclusions.

(1) Most figures lack a control condition, which impedes to obtain compelling conclusions.

Answer: We have added proper controls in all figures in the revised manuscript without changing the results and conclusions. Specifically, Mock condition was added as control to Fig 5A, 6E-6G and EV3F-3G, IgG was added as control to Fig 5E-5H.

(2) Previous studies (doi: 10.1371/journal.ppat.1009530 and doi: <https://doi.org/10.1128/spectrum.01098-22>) already reported the relevance of the DDX5-METTL3-METTL14 complex on the Type-I IFN and pro-inflammatory responses during VSV or IAV replication in MEFs and A549 cells, respectively. This, through the m6A-mediated regulation of DHX58, p65, IKK gamma, IFN beta and/or IL-6 mRNAs. In this sense, the m6A-mediated regulation of TLR2 and TLR4 mRNAs by the DDX5-METTL3-METTL14 complex during bacterial infection and their impact on IL-6 levels reported in this study is not surprising.

Note: *DDX5 interacted METTL14 or METTL3 also been discovered by Dattilo D et al. in the latest study (Dattilo D, Di Timoteo G, Setti A, et al. The m6A reader YTHDC1 and the RNA helicase DDX5 control the production of rhabdomyosarcoma-enriched circRNAs. Nat Commun. 2023;14(1):1898.).*

Answer: Although our previous study found that DDX5 regulates m6A modification during antiviral innate immunity, no studies are available about how RNA helicases contribute to bacterial infection and the homeostasis of inflammation. As viruses and bacteria are sensed by different TLRs, the down-stream signaling pathways may vary significantly and understanding how m6A modification modulates bacteria-induced inflammatory responses is essential. We think our findings are different from the study mentioned above in the following aspects:

(1) We have demonstrated that DDX5 interacted METTL14 to form the DDX5/METTL3/METTL14 complex, which was disturbed upon bacterial infection via the degradation of DDX5 by Hrd1-mediated ubiquitin proteasome pathway, our previous study proved DDX5 interacted with METTL3, while not found DDX5 also interacted with METTL14.

(2) We found that DDX5 specifically regulated m6A modification of TLR4/2 mRNAs to regulate TLR2/4-mediated inflammatory responses upon bacterial infection, providing a new mechanism to understand the transcription and translation of TLR4/2 mRNAs during bacterial infection, while DDX5 promotes viral infection via regulating N6-methyladenosine levels on the DHX58 and NFkB transcripts.

Therefore, our research findings provide vital mechanisms of bacterial infection and homeostasis of inflammation, which were not reported in any previous studies. We have stated this novelty in Introduction, Results and Discussion of the revised version.

Specific comments:

- Since data on Fig. 1 are mostly based on the analysis of IL-6 (secreted levels and promoter induction) in cell culture, it would be important to determine whether these observations are exclusive for this cytokine (or its mRNA) or whether also applies to other pro-inflammatory cytokine markers. This to decipher whether the effect of DDX5-METTL3-METTL14 complex is exerted on the IL-6 mRNA via m6A-YTHDF2 (as previously reported; doi: <https://doi.org/10.1128/spectrum.01098-22>) or on other players in the NF-kB pathway.

Answers: We agree with the reviewer's suggestion. We have performed new experiments to detect TNF- α production in DDX5^{+/+} BMDM, DDX5^{-/-} BMDM, MEFs or macrophages (M ϕ) after LPS, FSL-1, or Pam3CSK4 treatment, as well as in mice after bacterial infection. We have added the results in Fig 1B, 1D and 1E, Fig EV3J and Fig 7 E, 7H and 7J in the revised manuscript. Our results demonstrated that TNF-A was also inhibited by DDX5.

- From Fig. 2, it is unclear how TLR2 and TLR4 activation (by agonists or infection) results in reduced levels of DDX5 and how this relates to Hrd1-mediated ubiquitination and degradation of DDX5. Is DDX5 relocated to the ER or is Hrd1 relocated to the nucleus during bacterial infection and treatment with TLR2 and TLR4 agonists?

Answers: We agree with the reviewer's suggestion. We have performed new experiments to explore the mechanism of Hrd1-mediated degradation of DDX5. First, DDX5 was exported from the nucleus after the treatment with TLR2/TLR4 agonists in both mouse macrophages or MEFs (Fig 3A and 3B). Then, CLSM results showed the increased co-localization of DDX5 and Hrd1 in mouse macrophages upon stimulation with TLR2/TLR4 agonists (Fig 3I). We have added the results in Fig 3A, 3B and 3I in the revised manuscript.

- In my opinion, the most critical weakness of this work is the lack of a "Mock condition" (vehicle for the drugs) in all the experiments in which TLR2 and TLR4 agonists were used. Indeed, authors included this critical control only in some of them. This is critical because authors show that LPS, FSL-1 and Pam3CSK4 induce an important reduction of DDX5 and thus, all the observations and conclusions are performed under reduced levels of DDX5 that in some cases are further reduced by siRNA. See Figs. 3K, 3L, 4A-4K, 6A-6Q as examples.

Answers: We appreciate the reviewer's suggestion and added appropriate control or mock conditions in all figures of the revised manuscript, including Figs. Fig 5A, 6E-6G and EV3F-3G.

- From Fig. 4, if authors propose that DDX5 is required for the stability of the METTL3/METTL14 complex, why DDX5 knock down only affects the levels of m6A on the TLR2 and TLR4 mRNAs? Once again, in the absence of the vehicle control, all these conclusions

arise from data obtained under reduced levels of DDX5 because the treatment with LPS, FSL-1 and Pam3CSK4 (used as a control) that were further decreased by siRNA.

Answers: We agree with the reviewer that DDX5 knock down could affect m6A modification of many mRNAs ($\log_{2}FC \geq 2.5$). We have added this information into Fig EV3. We have followed the referee's advice to add the controls in the revised Fig EV3A-3C.

- What was the m6A pattern of TLR4 mRNA under LPS treatment and in the TLR2 mRNA under FSL-1 and Pam3CSK4 treatments?

Answers: The m6A pattern was located in the PR_intron of TLR4 mRNA when MEFs were treated with LPS, while the m6A pattern was located in the CDS of TLR2 mRNA when MEFs treated with FSL-1 and Pam3CSK4. We have added this information in the revised version.

- Fig. 4A: why the m6A site in the LPS-NC condition has A/C and not only A as the others?

Answers: We apologize for this mistake. We put a wrong picture in panel A of Figure 4 (LPS treated NC group). "RRACH" (R = G or A, H = A, C, or U) is the character of mammal mRNA m6A modification. We have replaced it with the right picture (GGACU, P-value ($1e-125$) in Fig 5C in the revised version.

- The sub-title "DDX5 halts translation of TLR2/4 transcripts..." should be corrected as authors do not evaluate translation in their experiments.

Answers: We thank the reviewer's advice. We have changed the sub-title to " DDX5 restrained inflammatory responses via regulating m6A modification of TLR2/4 transcripts during bacterial infection. " in the revised version.

Dear Prof. Chu

Thank you for the submission of your revised manuscript to our editorial offices. I have now received the reports from the three referees that I asked to re-evaluate your study, you will find below. As you will see, the referees now fully support the publication of the study in EMBO reports.

Before I can proceed with formal acceptance, I have these editorial requests I ask you to address in a final revised manuscript:

- I would suggest this modified title:
DDX5 inhibits inflammation by modulating m6A levels of TLR2/4 transcripts during bacterial infection
- Please have the entire final manuscript carefully proofread by a native speaker.
- Please provide the abstract written in present tense.
- Please upload a complete author checklist, which you can download from our author guidelines (<https://www.embopress.org/page/journal/14693178/authorguide>) with your final revised files. Please insert page numbers in the checklist to indicate where the requested information can be found in the manuscript. The completed author checklist will also be part of the RPF.

Please also follow our guidelines for the use of living organisms, and the respective reporting guidelines:
<http://www.embopress.org/page/journal/14693178/authorguide#livingorganisms>

- Please make sure that all the funding information is also entered into the online submission system and that it is complete and similar to the one in the acknowledgement section of the manuscript text file. Presently, the grant 'LVRI Yuan-Heng Talent Program (NKLS2020-119)' is only mentioned in the acknowledgements. Please check.
- Please move all the methods information and the additional references to the main manuscript text file and delete the file 'additional supplementary material'. We do not allow methods information in the Appendix.
- Please make sure that the number "n" for how many independent experiments were performed, their nature (biological versus technical replicates), the bars and error bars (e.g. SEM, SD) and the test used to calculate p-values is indicated in the respective figure legends (for main, EV and Appendix figures) of the final revised manuscript. Please also check that all the p-values are explained in the legend, and that these fit to those shown in the figure. Please provide statistical testing where applicable. Please avoid the phrase 'independent experiment', but clearly state if these were biological or technical replicates. Please also indicate (e.g. with n.s.) if testing was performed, but the differences are not significant. In case n=2, please show the data as separate datapoints without error bars and statistics. See also:
<http://www.embopress.org/page/journal/14693178/authorguide#statisticalanalysis>

If n<5, please show single datapoints for diagrams. In particular:

Please note that for the figures EV5a-b, p-values and statistical tests are indicated in the legends. However, "*"/** is not been represented in the figures. Please rectify this in the figures or legends as applicable.

- Please add to each legend a 'Data Information' section explaining the statistics used or providing information regarding replicates and scales. See:

- Please add scale bars of similar style and thickness to the microscopic images (main and EV figures), using clearly visible black or white bars (depending on the background). Please place these in the lower right corner of the images themselves. Please do not write on or near the bars in the image but define the size in the respective figure legend.
- Please remove the reagents and tools table from the main manuscript text file. I have attached templates for that in word or excel format. Please upload the filled in table to the manuscript tracking system as 'Reagent Table' file. Please also adjust any callouts to this table. The example linked below shows how the table will display in the published article and includes examples of the type of information that should be provided for the different categories of reagents and tools. Please list your reagents/tools using the categories provided in the template and do not add additional subheadings to the table. Reagents/tools that do not fit in any of the specific categories can be listed under "Other":
https://www.embopress.org/pb%2Dassets/embo-site/msb_177951_sample_FINAL.pdf
- The Data Availability section should only contain information on large datasets that have been deposited to external

repositories and all access information. If no primary datasets have been deposited, please state this here, e.g. using 'No primary datasets have been generated and deposited'. Please remove the sentence 'All study data are included in the main text, Expanded View and Additional Supplementary material'.

- We now request the publication of original source data with the aim of making primary data more accessible and transparent to the reader. Our source data coordinator will contact you to discuss which figure panels we would need source data for and will also provide you with helpful tips on how to upload and organize the files.

Please order the manuscript sections like this, using these names:

Title page - Abstract - Keywords - Introduction - Results - Discussion - Materials and Methods - Data availability section - Acknowledgements - Disclosure and Competing Interests Statement - References - Figure legends - Expanded View Figure legends

In addition, I would need from you:

- a short, two-sentence summary of the manuscript (not more than 35 words).
- two to four short (!) bullet points highlighting the key findings of your study (two lines each).
- a schematic summary figure that provides a sketch of the major findings (not a data image) in jpeg or tiff format (with the exact width of 550 pixels and a height of not more than 400 pixels) that can be used as a visual synopsis on our website.

Best,

Referee #1:

The authors have addressed all critiques.

Referee #2:

The revised version has addressed my concerns. I have no further questions.

Referee #3:

Authors have satisfactorily answered to all my comments, particularly those related to the lack of controls in some experiments.

The authors addressed the minor editorial issues.

Dear Prof. Chu,

Thank you for the submission of your further revised manuscript to our editorial offices. I now went through the manuscript and note that the figures are much too crowded and do not have the right size requirements. In their present form, the figures (main and most of the EV figures) cannot be typeset in a reasonable way so that readers can see details when printing the pdf of the paper.

Thus, before I can proceed with formal acceptance, I have these further editorial requests:

- Please move a substantial part of the data now shown in the main and EV figures to an Appendix. For all the present figures, except EV4 and EV5, it will be necessary to move half the panels shown to the Appendix. Please select these panels yourself and create a single pdf file labeled Appendix. The Appendix should have page numbers and needs to include a table of content on the first page (with page numbers) and legends for all content. Please follow the nomenclature Appendix Figure Sx, Appendix Table Sx etc. throughout the text, and also label the figures and tables according to this nomenclature. Please also move the respective parts of the figure legends to the Appendix file and remove these from the main manuscript. Finally, please update all the callouts.

The remaining main and EV figures need to be formatted according to our guidelines:

- For the panels 3I, 4B and 4I it remains unclear from the legend what has been measured and what the diagrams next to the microscopic images show. What does the black line in the merged image indicate? Please provide a more detailed legend.

- Please also remove the source data (SD) for those panels that have been moved to the Appendix from the SD folders. Please upload the final SD for the main figures as one folder per figure.

- Please provide a synopsis image with bigger fonts. Presently, part of the writing is not legible. Please keep the final size of the image (exact width of 550 pixels and a height of not more than 400 pixels).

Please use this link to submit your revision: <https://embor.msubmit.net/cgi-bin/main.plex>

Best,

The authors addressed the remaining editorial issues.

Prof. Yue-Feng Chu
Lanzhou Veterinary Research Institute, Chinese Academy of Agricultural Sciences
No.1, Xujiaping, Chengguan District, Lanzhou, Gansu Province
Gansu Province 730046
China

Dear Prof. Chu,

I am very pleased to accept your manuscript for publication in the next available issue of EMBO reports. Thank you for your contribution to our journal.

Yours sincerely,
